# In vivo assembly enhanced binding effect augments tumor specific ferroptosis therapy

Da-Yong Hou [1,2,3,5], Dong-Bing Cheng [4,5], Ni-Yuan Zhang[1,5], Zhi-Jia Wang[1,2,3], Xing-Jie Hu[1], Xin Li[4], Mei-Yu Lv[2], Xiang-Peng Li[1,2,3], Ling-Rui Jian[2,3], Jin-Peng Ma[2,3], Taolei Sun [4] ✉, Zeng-Ying Qiao [1] ✉, Wanhai Xu [2,3] ✉ & Hao Wang [1] ✉

Emerging evidence indicates that the activation of ferroptosis by glutathione peroxidase 4 (GPX4) inhibitors may be a prominent therapeutic strategy for tumor suppression. However, the wide application of GPX4 inhibitors in tumor therapy is hampered due to poor tumor delivery efficacy and the nonspecific activation of ferroptosis. Taking advantage of in vivo self-assembly, we develop a peptide-ferriporphyrin conjugate with tumor microenvironment specific activation to improve tumor penetration, endocytosis and GPX4 inhibition, ultimately enhancing its anticancer activity via ferroptosis. Briefly, a GPX4 inhibitory peptide is conjugated with an assembled peptide linker decorated with a pH-sensitive moiety and ferriporphyrin to produce the peptide-ferriporphyrin conjugate (**Gi-F-CAA**). Under the acidic micro-environment of the tumor, the **Gi-F-CAA** self-assembles into large nano-particles (Gi-F) due to enhanced hydrophobic interaction after hydrolysis of CAA, improving tumor endocytosis efficiency. Importantly, Gi-F exhibits sub-stantial inhibition of GPX4 activity by assembly enhanced binding (**AEB**) effect, augmenting the oxidative stress of ferriporphyrin-based Fenton reaction, ultimately enabling antitumor properties in multiple tumor models. Our findings suggest that this peptide-ferriporphyrin conjugate design with **AEB** effect can improve the therapeutic effect via induction of ferroptosis, providing an alternative strategy for overcoming chemoresistance.

Ferroptosis, as an iron-dependent and non-apoptotic form of programmed cell death, is characterized by aberrant elevation of iron-mediated oxidative stress and exhaustion of glutathione (GSH) which induces excessive accumulation of lethal lipid peroxides (LPO)[1,2]. With the deepened comprehension of molecular mechanism, glutathione peroxidase 4 (GPX4) has been thoroughly reported as a key regulator for protecting cells from toxic lipid hydroperoxides of ferroptosis, which is highly expressed in tumor tissues compared with that in adjacent normal tissues[3–5]. Emerging evidences have indicated that the activation of ferroptosis by GPX4 inhibitors (such as peptide GACNWLPLYPCPV[6], derived by screening peptide libraries displayed on T7 phages) is increasingly recognized as a prominent therapeutic strategy for tumor suppression[7,8]. However, the wide application of GPX4 inhibitor in tumor therapy is hindered due to the poor tumor delivery efficacy and nonspecific activation of ferroptosis[9–11].

[1]CAS Key Laboratory for Biomedical Effects of Nanomaterials and Nanosafety, CAS Center for Excellence in Nanoscience, National Center for Nanoscience and Technology (NCNST), Beijing 100190, China. [2]NHC and CAMS Key Laboratory of Molecular Probe and Targeted Theranostics, Harbin Medical University, Harbin 150001, China. [3]Department of Urology, Harbin Medical University Cancer Hospital, Heilongjiang Key Laboratory of Scientific Research in Urology, Harbin 150001, China. [4]School of Chemistry, Chemical Engineering & Life Science, Hubei Key Laboratory of Nanomedicine for Neurodegenerative Diseases, Wuhan University of Technology, No. 122 Luoshi Road, Wuhan 430070, PR China. [5]These authors contributed equally: Da-Yong Hou, Dong-Bing Cheng, Ni-Yuan Zhang. ✉e-mail: suntl@whut.edu.cn; qiaozy@nanoctr.cn; xuwanhai@hrbmu.edu.cn; wanghao@nanoctr.cn

Nanomaterials have been demonstrated to improve the performance of traditional chemotherapeutic agents or targeted drugs by compositing multiple functional agents. Whereas, the infiltration of nanomaterials into tumor tissue as well as distal tumor cells from blood vessels remains an unresolved obstacle owing to the limited extravasation, elevated tumor interstitial fluid pressure, tightly packed tumor cells and dense extracellular matrix[12]. With advantages of small molecule and nanoscience, advanced bionanomaterials in a more controllable manner can enhance the penetration, accumulation and cellular internalization of cytotoxic agent in tumor tissues while sparing the normal tissues and metabolic organs[13–16]. On this basis, our group developed an in vivo self-assembly strategy[17–19] based on supramolecular chemistry, which has been widely recognized as a promising tool for enhancing the accumulation of cytotoxic agent to its action sites by overcoming the physical or metaphorical barriers. For instance, the in situ constructed fibrous nanomaterials was reported to obviously enhance the mitochondria destruction by multivalent cooperative interactions[20]. Additionally, the nanomaterials based on enzyme-activated self-assembly strategy was developed for precise drug delivery, which exhibited high penetration depth and improved accumulation in tumor[21,22].

In this work, we develop a peptide-ferriporphyrin conjugate with tumour microenvironment specific activation for improved tumor penetration and endocytosis, which enables effective inhibition of GPX4 due to the assembly enhanced binding (AEB) effect, ultimately improving its anticancer activity via ferroptosis (Fig. 1).

Briefly, a GPX4 inhibitory peptide (GACNWLPLYPCPV)[6] is conjugated with an assembly peptide linker (LKLKLK)[23,24] decorated with a pH-sensitive moiety (cis-aconitic anhydride, CAA)[25] and ferriporphyrin (FeTCPP) to produce the peptide-ferriporphyrin conjugate (**Gi-F-CAA**, FeTCPP-LKLKLK(CAA)GACNWLPLYPCPV). Owing to the improved hydrophilicity after CAA modification, **Gi-F-CAA** in single chain state with a small size penetrates deeply into solid tumors. Under the acidic microenvironment of the tumor, the hydrolysis of CAA induces self-assembly of **Gi-F-CAA** into large nanoparticles (Gi-F) due to the enhanced hydrophobic interaction. In turn, this increases tumor endocytosis efficiency and subsequent tumor accumulation. Moreover, due to augmentation of GPX4 binding sites provided by the formed nanostructure, Gi-F exhibits increased inhibition of GPX4 activity by assembly enhanced binding (**AEB**) effect in cells, increasing the reactive oxidative species (ROS) produced by FeTCPP-based Fenton reaction. Together, this elicits anti-tumour effects via induction of ferroptosis. Importantly, Gi-F-CAA exhibits potential in several tumor models including bladder cancer, multi-drug resistant (MDR) breast cancer and large renal cell carcinoma. Furthermore, this peptide-ferriporphyrin conjugates design could be applied for delivery of various functional agents, such as chemotherapeutics, peptide drugs and fluorescence contrast agents. Taken together, we anticipate that the development of the concept described above may improve tumor specific delivery of functional agents while enabling function enhancement due to its convenient modular molecular modification.

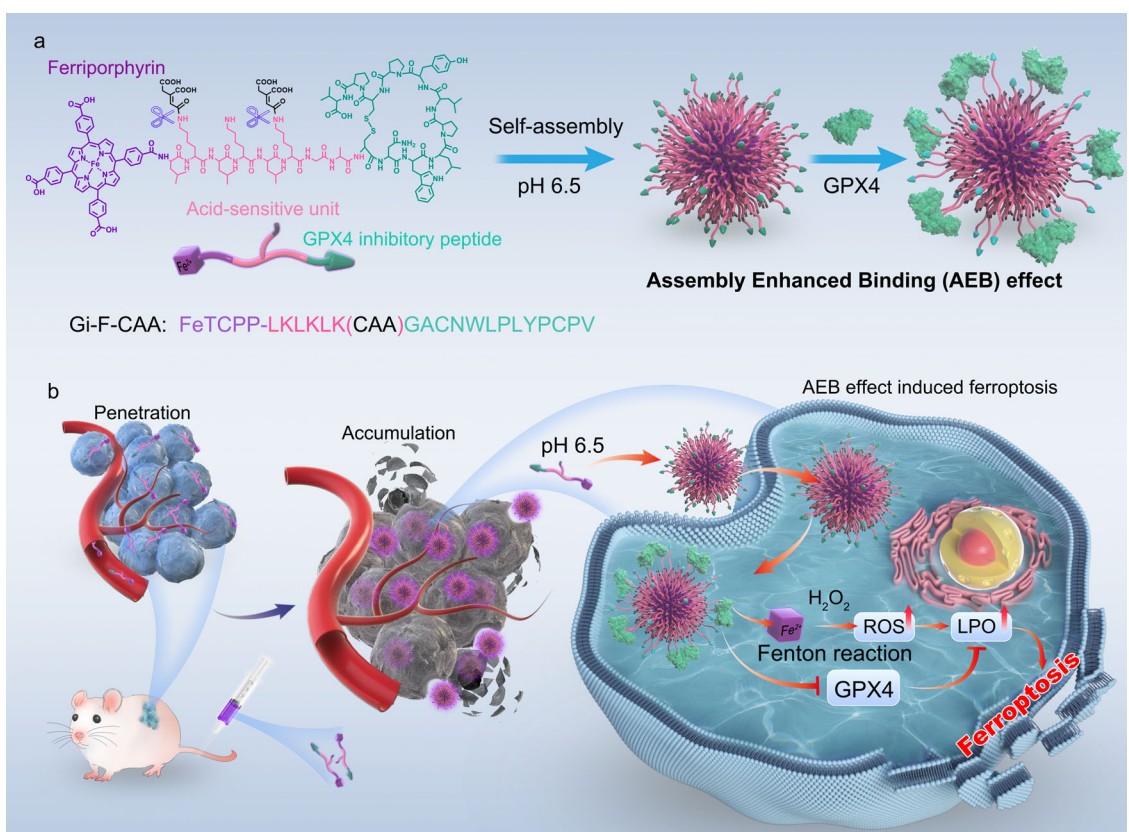

**Fig. 1 | A tumor microenvironment specifically activated peptide-ferriporphyrin conjugates** (Gi-F-CAA) **for ferroptosis therapy of bladder cancer by assembly enhanced binding effect. a** Molecular structure and acid-responsive self-assembly behavior of **Gi-F-CAA**. Briefly, a GPX4 inhibitory peptide (GACNWL-PLYPCPV), a peptide linker (LKLKLK) decorated with a pH-sensitive moiety (cis-aconitic anhydride, CAA) and ferriporphyrin (FeTCPP) were conjugated to produce the peptide-ferriporphyrin conjugates (**Gi-F-CAA**, FeTCPP-LKLKLK(CAA) GACNWLPLYPCPV). **b Gi-F-CAA** with a small size could easily penetrate deeply into solid tumor, which self-assembled into large nanoparticles under the acidic tumor microenvironment to improve the tumor endocytosis efficiency. Importantly, substantial inhibition of GPX4 activity was achieved by assembly enhanced binding (**AEB**) effect, which augmented the oxidative stress of FeTCPP-based Fenton reaction, ultimately enabling the remarkable antitumor properties by ferroptosis.

## Results

### Molecular construction and acid-responsive self-assembly behavior

The peptide-porphyrin conjugates TCPP-LKLKLKGACNWLPLYPCPV was prepared using a standard solid-phase peptide synthesis method followed by oxidation in low concentration and purified by reverse phase high-performance liquid chromatography (Supplementary Figs. 1 and 2). Afterwards, the acid-responsive molecule peptide-ferriporphyrin conjugates **Gi-F-CAA** (FeTCPP-LKLKLK(CAA)GACNWL-PLYPCPV) was obtained through iron chelate of TCPP-LKLKLKGACNWLPLYPCPV and subsequent decoration with CAA (Supplementary Figs. 3 and 4). The two characteristic Q-band peaks in UV-visible absorption spectrum and the appearance of CAA signals in [1]H NMR spectrum confirmed the structure of **Gi-F-CAA**. The control peptide-porphyrin conjugates, including Gi-F-SA decorated with non-sensitive succinic anhydride (SA) moieties (FeTCPP-LKLKLK(SA) GACNWLPLYPCPV), NGi-F-CAA without GPX4 inhibition ability (FeTCPP-LKLKLK(CAA)GACLNWPYLPCPV), and Gi-F with nanoparticle forming ability (FeTCPP-LKLKLKGACNWLPLYPCPV), were prepared by the similar method (Supplementary Figs. 5–7).

Two single peptide structures were depicted in Fig. 2a, b using MD simulations. Both Gi-F and **Gi-F-CAA** could fold a turn structure due to the existence of S-S bond formed by two cysteines. Besides, the positively charged lysines in Gi-F and the negatively charged CAA-Lys in **Gi-F-CAA** were all exposed to the solvent. This charge differences affect the aggregation of these two kinds of peptides. The positively charged lysine residues neutralized the negative charge of carboxyl group in the porphyrin part, causing the aggregation of the Gi-F. While the negatively charged **Gi-F-CAA** made the overall structure even more negative, preventing the closure of CAA-Lys residues and porphyrin part, which were more stable in single chain state. As shown in Fig. 2c, d, the Gi-F chains could get together spherically within 200 ns to form a nanoparticle in which the porphyrin parts were distributed in the inside and other parts in the outside, while the **Gi-F-CAA** chains became apart from each other even though the initial structure was set to be a nanosphere with the porphyrin parts distributed in the inside of the nanosphere (Supplementary Fig. 8). This could also be verified by the solvent accessible surface area (SASA) value of the two systems. As shown in Supplementary Fig. 9, the SASA values of Gi-F system were around 500 Å$^2$ during MD simulations while the corresponding values in **Gi-F-CAA** system were >750 Å$^2$ during MD simulations.

Moreover, the mechanism of mild acid induced self-assembly was studied by hydrolysis process of CAA and ζ-potential change. As shown in Fig. 2e and Supplementary Fig. 10, compared with that at pH 7.4, the CAA hydrolysis rate of **Gi-F-CAA** at pH 6.5 significant increased. As control group, the acid-nonsensitive Gi-F-SA showed little hydrolysis both at pH 6.5 and 7.4, suggesting the Gi-F-SA could maintain stable structure during incubation in mildly acidic condition (Supplementary Fig. 11). Similarly, the ζ-potential increased considerably after **Gi-F-CAA** incubated at pH 6.5 (Supplementary Figs. 12 and 13), which further proved the CAA hydrolysis. Both the results indicated the enhanced hydrophobicity and surface charge change resulting from CAA hydrolysis were the main drive forces for in vivo self-assembly. The critical aggregation concentrations (CAC) were applied for investigating the assembly ability of peptide-porphyrin conjugates. It should be noted that the CAC of Gi-F was 13.8 μM in phosphate buffer solution (PBS, 0.01 M, pH 7.4). After the decoration with CAA, the **Gi-F-CAA** possessed the CAC of 149.5 μM, which demonstrated **Gi-F-CAA** remained as single chains and Gi-F self-assembled into nanoaggregates at the concentration between 13.8 and 149.5 μM (Fig. 2f). The self-assembly behavior of Gi-F, Gi-F-SA, NGi-F-CAA and **Gi-F-CAA** responding to acid was explored by dynamic light scattering (DLS) and transmission electron microscopy (TEM) (Supplementary Figs. 14–16). The DLS revealed the particles sizes of **Gi-F-CAA** increase from 13 ± 3 nm to 78 ± 11 nm after incubation at pH 6.5 for 1 h, whereas it

exhibited no obvious changes at pH 7.4 (Fig. 2g). After incubation with PBS (0.01 M, pH 6.5) for 1 h, **Gi-F-CAA** displayed an obvious particle morphology with size of 57 ± 9 nm, suggesting the mild acid could realize the self-assembly (Fig. 2h and Supplementary Fig. 17).

### AEB effect of Gi-F-CAA

The protein docking process was applied for the nanoparticle to dock to GPX4mu protein (PDB ID: 5H5Q) after confirming the assembly of Gi-F. And then 200 ns long time MD simulations were carried out to relax the binding between Gi-F nanoparticle and GPX4mu protein. Owing to the much bigger size of Gi-F nanoparticle than GPX4mu protein, it was possible for multiple GPX4mu proteins to bind to Gi-F nanoparticle simultaneously (Fig. 2i). Here only one GPX4mu protein was deployed to bind with the Gi-F nanoparticle to simplify the system as shown in Fig. 2i. According to the binding structure shown in Fig. 2j, there were on average ~2 peptides interacting with one GPX4mu protein. Meanwhile, the binding energy between the Gi-F nanoparticle and the GPX4mu protein was also calculated. It turned out that the van der Waals interaction played an important role for the binding between Gi-F nanoparticle and GPX4mu protein, since the value of van der Waals interaction was as large as 60 kJ/mol, which was comparable to the electrostatic interaction (99 kJ/mol). In addition, the GPX4 protein was observed to be distributed on the surface of Gi-F nanoparticle based on the TEM images, indicating Gi-F could bind with GPX4 effectively (Fig. 2k). The DLS revealed the particles sizes of Gi-F was ~90 ± 9 nm after incubation with GPX4 (Supplementary Fig. 18). Furthermore, the Hill plot was introduced to describe the multiple binding patterns between Gi-F nanoparticle and GPX4. As shown in Fig. 2l, the Job plot analysis of fluorescent intensity versus concentration fit well with Hill plot. As a result, Gi-F nanoparticle exhibited a higher Hill coefficient ($n = 2.1$) toward GPX4 compared with that of **Gi-F-CAA** single chain ($n = 1.1$), suggesting the stronger multivalent cooperative interactions and higher binding affinity between nanoparticle and GPX4 (Fig. 2l). Apparently, the nanoparticles held larger contact area and more interaction sites with GPX4 than single chain, showing noteworthy **AEB** effect. Subsequently, Microscale Thermophoresis (MST) ligand binding experiment was performed to determine the binding affinity of **Gi-F-CAA**, Gi-F and NGi-F-CAA to GPX4 (Fig. 2m and n). As expected, NGi-F-CAA exhibited an extremely decreased binding affinity to GPX4 (apparent K$_d$: ~17.62 μM) owing to the disordered sequence of GPX4 inhibitory peptide (Supplementary Fig. 19). However, Gi-F nanoparticle (Fig. 2n) showed a ~ 55 times increased binding affinity to GPX4 (apparent K$_d$: ~30.29 nM) by the multivalent cooperative interactions than that of **Gi-F-CAA** (K$_d$: ~1.68 μM) (Fig. 2m) demonstrating that the in vivo self-assembled nanostructure obviously enhanced selectivity toward GPX4 with higher binding affinity.

In order to investigate the catalytic activity of ferriporphyrin based on Fenton reactions, the production of hydroxyl radical (•OH) was measured by electron paramagnetic resonance (EPR) and fluorescence spectra. 5,5-dimethyl-1-pyrroline-N-oxide (DMPO) was applied as radical trap, and the characteristic 1:2:2:1 quartet signal was detected after **Gi-F-CAA** treated with 1 mM H$_2$O$_2$ in the EPR spectrum, which demonstrated the generation of •OH (Supplementary Fig. 20). In addition, the similar signal intensity between Gi-F and **Gi-F-CAA** indicated the self-assembly behavior had little impact on the catalytic activity of ferriporphyrin. Terephthalic acid (TPA) had been approved as a specific and sensitive probe for •OH monitoring, which was a nonfluorescent molecule and could quantitatively react with •OH to generate a fluorescent product hydroxylated terephthalic acid (HTPA)[26], The H$_2$O$_2$ concentration-dependent variation of TPA fluorescence intensity showed a linear relationship, and the calculated turnover frequency (TOF) of Gi-F was $2.2 \times 10^{-2}$ s$^{-1}$, which was similar to that of conventional Fenton system ($1.5 \times 10^{-2}$ s$^{-1}$)[27], suggesting the appropriate catalytic activity for ferroptosis in cancer treatment (Supplementary Fig. 21). It was reported that the iron could convert

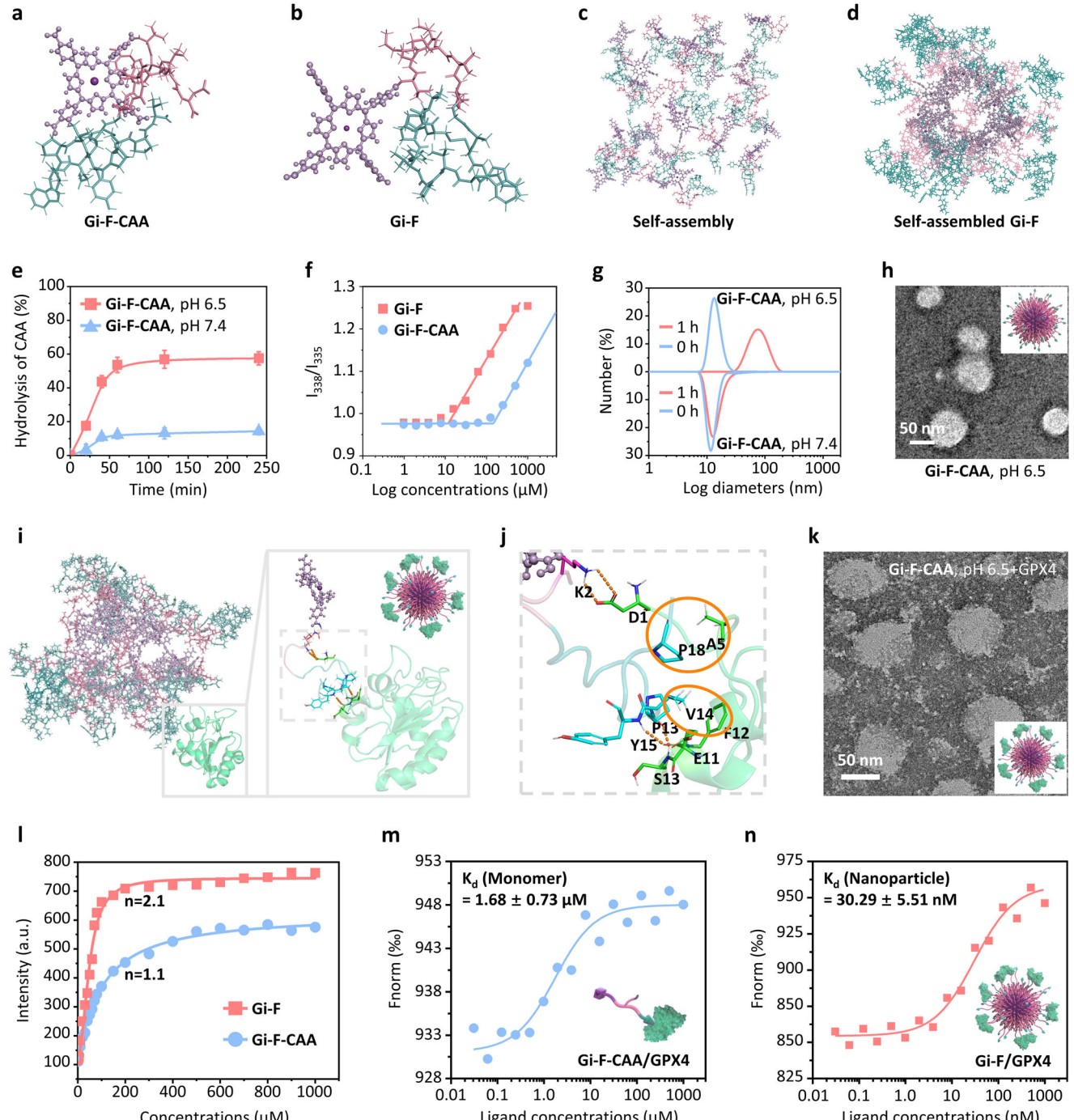

**Fig. 2 | Assembly enhanced binding (AEB) effect of Gi-F-CAA. a** The structure of **Gi-F-CAA** (peptide-porphyrin conjugate decorated with acid-sensitive CAA moieties). **b** The structure of Gi-F (peptide-porphyrin conjugate with nanoparticle forming ability). The porphyrin parts were shown in sphere-stick mode with the carbon atoms colored in green and the other parts were shown in stick mode with the carbon atoms colored in cyan. **c** The self-assembly progress of Gi-F. The porphyrin parts were shown in sphere-stick mode with the carbon atoms colored in green and the other parts were shown in stick mode with the carbon atoms colored in cyan. The MD simulations begun from an initial randomly distributed system. **d** The self-assembled nanoparticle of Gi-F. **e** The hydrolysis profiles of **Gi-F-CAA** at pH 7.4 and 6.5 (PBS, 0.01 M) measured by HPLC ($n = 3$ experimental repeats). Experiment was independently repeated three times with similar results. **f** CAC values determined by $I_{338}/I_{335}$ ratio from pyrene as a function of concentration of **Gi-F-CAA** and Gi-F. **g** Particle size change of **Gi-F-CAA** and Gi-F (40 μM) in PBS (0.01 M, pH 7.4 or 6.5) measured by DLS. **h** Representative TEM images of **Gi-F-CAA**

in PBS at pH 7.4 for 1 h ($n = 3$ experimental repeats). Experiment was independently repeated three times with similar results. **i** The binding mode between self-assembled nanoparticle and the GPX4mu protein. The porphyrin parts were shown in sphere-stick mode with the carbon atoms colored in green and the other parts were shown in stick mode with the carbon atoms colored in cyan. The GPX4mu protein was shown in cartoon mode and colored in magenta. **j** Key residues between self-assembled nanoparticle and the GPX4mu protein. **k** Corresponding TEM images of **Gi-F-CAA** at pH 6.5 incubated with protein GPX4. **l** Affinity binding curve of **Gi-F-CAA** and Gi-F with GPX4. n represents the Hill coefficient calculated from Hill plot. **m** The binding affinity of **Gi-F-CAA** to GPX4 by Microscale Thermophoresis (MST) ligand binding measurements. $K_d$: the dissociation constant. **n** The binding affinity of Gi-F to GPX4 by MST ligand binding measurements. $K_d$: the dissociation constant. Data were expressed as mean ± SD. Source data are provided as a Source Data file.

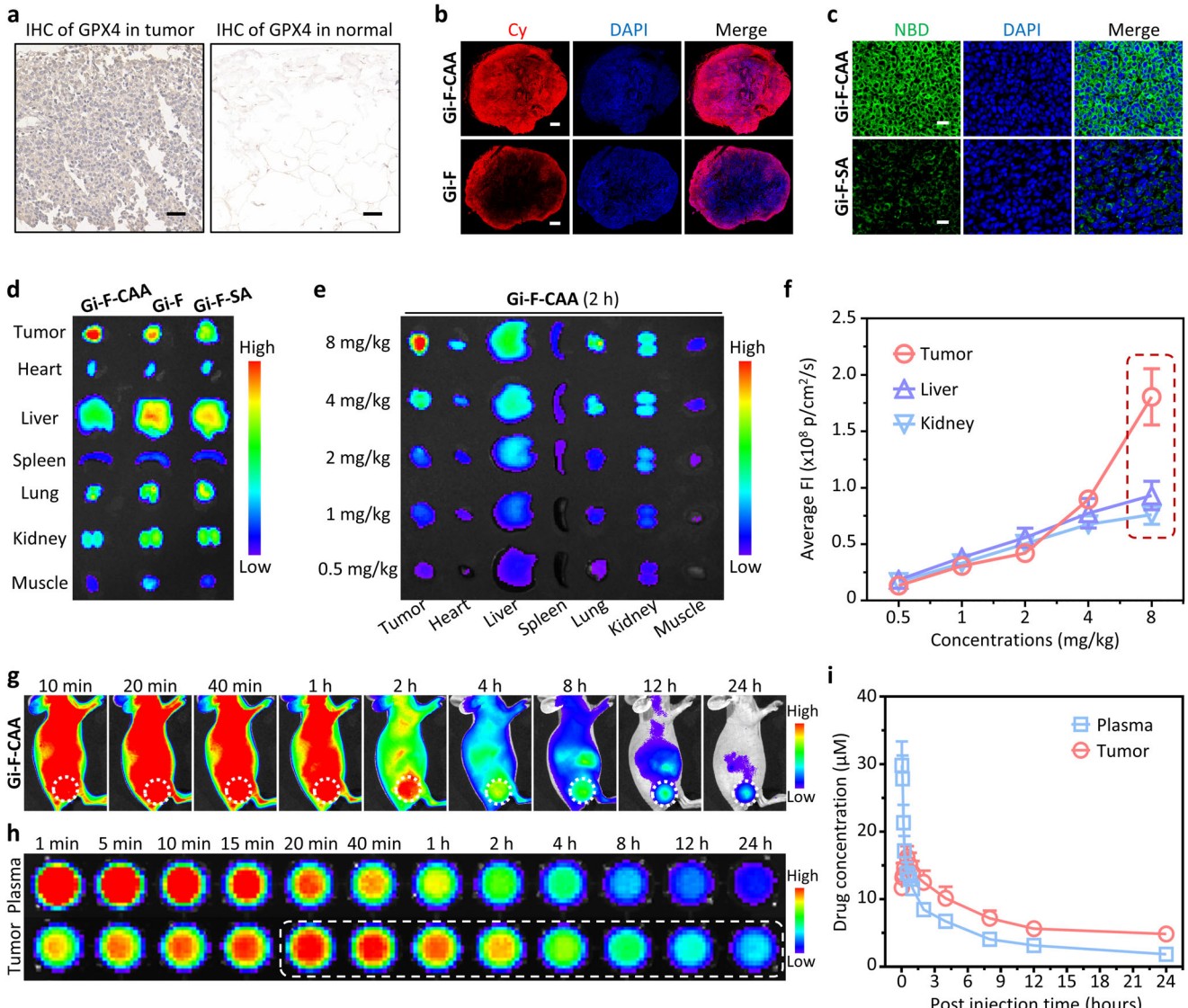

**Fig. 3 | In vivo self-assembly induced tumor accumulation of Gi-F-CAA. a** GPX4 immunohistochemical staining of bladder tumor tissues and normal bladder tissues. Scale bars: 50 μm. **b** Fluorescence of images of the whole tumor tissues after treated with Cy labeled **Gi-F-CAA** and Gi-F (8 mg/kg in 100 μL PBS). Scale bars: 1 mm. **c** Fluorescence images of tumor tissues after treated with NBD labeled **Gi-F-CAA** and Gi-F-SA (8 mg/kg in 100 μL PBS). Scale bars: 20 μm. **d** Representative ex vivo fluorescence images of tumor and major organs (heart, liver, spleen, lung, kidney and muscle) collected after intravenous administrated with **Gi-F-CAA**, Gi-F and Gi-F-SA (8 mg/kg in 100 μL PBS), respectively (*n* = 3 mice). **e** Dose-dependent ex vivo fluorescence images of tumor and major organs after intravenous administrated with **Gi-F-CAA** (8 mg/kg, 4 mg/kg, 2 mg/kg, 1 mg/kg and 0.5 mg/kg in 100 μL PBS). **f** Corresponding quantitative fluorescence intensity of tumor, liver and kidney after intravenous administrated with **Gi-F-CAA** (*n* = 3 mice). **g** Representative time-dependent in vitro fluorescence images of EJ xenograft mice after intravenous administrated with **Gi-F-CAA** (8 mg/kg in 100 μL PBS) (*n* = 3 mice). **h** Representative time-dependent fluorescence images of tumor tissue lysates and plasma after intravenous administrated with **Gi-F-CAA** (8 mg/kg in 100 μL PBS) (*n* = 3 mice). **i** Corresponding quantitative fluorescence intensity of tumor tissue lysates and plasma after intravenous administrated with **Gi-F-CAA** (8 mg/kg in 100 μL PBS) (*n* = 3 mice). Data were expressed as mean ± SD. Source data are provided as a Source Data file.

intracellular reductive molecule GSH to GSSG to impair the oxidation resistance ability of cancer cells[28], thus the GSH depletion and •OH generation of peptide-porphyrin conjugates were assessed separately under $H_2O_2$ treatment (Supplementary Fig. 22). The similar relative GSH concentration depletion and •OH generation between **Gi-F-CAA** and Gi-F further confirmed the consistent catalytic activity of ferriporphyrin, which speculated acid-activated in vivo self-assembly could realize ferroptosis at the cell and animal level.

**In vivo self-assembly induced tumor accumulation of Gi-F-CAA**
GPX4, as a major lipid peroxidation scavenger, has been confirmed to be overexpressed in a variety of tumors, which plays a critical role in protecting tumor cells from ferroptosis[4,29]. Therefore,

immunohistochemistry (IHC) was performed to investigate the expression of GPX4 in bladder tumor tissues from 36 patients combined with normal bladder tissues from 18 patients. After obtaining appropriate informed consent, fresh bladder cancer specimens as well as normal bladder urothelium specimens were collected and fixed for analysis. The results suggested GPX4 was highly overexpressed in bladder tumor tissues compared with that in normal tissues (Fig. 3a and Supplementary Fig. 23). Taken together, these results demonstrated that GPX4 hold great potential in developing tumor specific controllable nanomaterials against bladder cancer.

Initially, the high tumor penetrability of **Gi-F-CAA** in vivo was verified with frozen sections of solid tumors. The tumor tissues of EJ xenograft mice were collected for fluorescence imaging after

administered with Cy-labeled **Gi-F-CAA** (8 mg/kg in 100 μL PBS). As a result, the fluorescence of Gi-F was observed to be only distributed around the periphery of tumor tissues (Fig. 3b). The fluorescence signal of Gi-F-SA was found to penetrate into the tumor interior, while in a week fluorescence intensity due to lacking of assembly ability (Supplementary Fig. 24a). In comparison, the fluorescence of **Gi-F-CAA** was observed to be distributed all over the tumor tissues, indicating the deep penetration ability of **Gi-F-CAA**. 4-nitro-2,1,3-benzoxadiazole (NBD), as a hydrophobic environment responsive fluorophore, was widely used for the detection of nanofibrils formation[30]. Subsequently, the in situ self-assembly of **Gi-F-CAA** in tumor was investigated by injecting NBD-labeled **Gi-F-CAA** into EJ xenograft mice. As a result, obvious green fluorescence was observed in **Gi-F-CAA** treated tumor as well as the moderate green fluorescence in Gi-F treated tumor, while not in Gi-F-SA groups (Fig. 3c and Supplementary Fig. 24b), indicating the weakly acidic microenvironment triggered in situ self-assembly in tumor.

To investigate the in vivo self-assembly induced tumor accumulation, ex vivo fluorescence imaging of tumor and major organs (heart, liver, spleen, lung, kidney and muscle) was carried out after intravenous administrated with Cy-labeled **Gi-F-CAA**, Gi-F and Gi-F-SA (8 mg/kg in 100 μL PBS), respectively. As a result, strong fluorescence signal was observed at the tumor treated with **Gi-F-CAA**, which was significantly higher than that in Gi-F and Gi-F-SA groups, respectively (Fig. 3d). Moreover, ex vivo fluorescence imaging of tumor and major organs was performed to investigate the dose-dependent tumor accumulation of **Gi-F-CAA** (Fig. 3e and Supplementary Fig. 25). The fluorescence signals at liver or kidney linearly increased as the dose of **Gi-F-CAA** increased from 0.5 mg/kg to 8 mg/kg. Although the similar phenomena were observed for **Gi-F-CAA** from 0.5 mg/kg to 2 mg/kg, the fluorescence signals at the tumor site were exponentially augmented as the dose of **Gi-F-CAA** increases from 2 mg/kg to 8 mg/kg (Fig. 3f). Quantitatively, the fluorescence intensity at tumor was ~1.9 fold and ~2.4 fold higher than that in liver and kidney, respectively. These results suggested that when the concentration of **Gi-F-CAA** at tumor site was higher than CAC, tumor accumulation would be extremely increased owing to the in situ self-assembly into nanoparticles. Subsequently, time-dependent in vivo fluorescence imaging of EJ xenograft mice was conducted to investigate the in vivo biodistribution and metabolism of **Gi-F-CAA** after administrated with **Gi-F-CAA** (8 mg/kg in 100 μL PBS). As a consequence, **Gi-F-CAA** was observed to be continuously accumulated at the tumor site from 10 min to 40 min and slowly metabolized from 1 h to 24 h (Fig. 3g). Representative time-dependent fluorescence images of tumor tissue lysates and plasma were acquired after intravenous administration with **Gi-F-CAA** (8 mg/kg in 100 μL PBS) (Fig. 3h). As a result, the fluorescence signals of plasma were rapidly decreased from 1 min to 20 min with a blood circulation half-life ($t_{1/2}$) of $62.0 \pm 11.5$ min, indicating the rapid plasma elimination behavior of **Gi-F-CAA**. In contrast, the fluorescence signals of tumor tissue lysates rapidly increased from 1 min to 40 min, suggesting the rapid in vivo tumor accumulation behavior of **Gi-F-CAA**, which were about 3-fold higher than that of plasma from 1 h to 24 h (Fig. 3i). Moreover, the concentration of **Gi-F-CAA** (16.2 μM) at tumor site was demonstrated to be higher than CAC (13.8 μM) based on the corresponding quantitative fluorescence intensity of tumor tissue lysates (Fig. 3i). These results further demonstrated that the **Gi-F-CAA** showed tremendous advantages in overcoming the bottleneck of tumor accumulation.

### In vitro enhanced anti-tumor ability of Gi-F-CAA based on ferroptosis

The cytotoxicity of **Gi-F-CAA** toward bladder cancer cells was investigated by CCK-8 assay (Fig. 4a). As a result, the half maximal inhibitory concentration values ($IC_{50}$) of Gi-F-SA exhibited a moderate decrease to 40.02 μM compared with NGi-F-CAA at pH 7.4 (154.80 μM) or NGi-F-

CAA at pH 6.5 (110.60 μM). Meanwhile, the $IC_{50}$ values of **Gi-F-CAA** (53.82 μM) were similar to that of Gi-F-SA at pH 7.4 (Fig. 4a). Whereas, the **Gi-F-CAA** exhibited an obviously enhanced cytotoxicity under mildly acidic conditions (pH 6.5) with $IC_{50}$ value of 14.28 μM, which might be attributed to improved internalization of self-assembled **Gi-F-CAA**. Moreover, similar results were observed in Gi-F at pH 7.4 with an obvious increased cytotoxicity (11.30 μM) (Fig. 4a). To demonstrate that the cytotoxicity caused by **Gi-F-CAA** was specifically related to ferroptosis, an inhibitor assay was performed with a ferroptosis inhibitor (Fer-1). As a result, the cytotoxicity of **Gi-F-CAA** was successfully rescued by Fer-1, indicating the ferroptosis-based tumor toxicity (Supplementary Fig. 26). Meanwhile, we compared **Gi-F-CAA** with Gi + FeTPP and Gi-CAA (intermediate of **Gi-F-CAA** without iron chelate, TCPP-LKLKLK(CAA)GACNWLPLYPCPV) by cell toxicity assay. As a result, Gi-F-CAA exhibited an obviously enhanced cytotoxicity than that of Gi + FeTPP, indicating the advantages of conjugation strategy. Besides, **Gi-F-CAA** exhibited an obviously enhanced cytotoxicity than that of Gi-CAA (Supplementary Fig. 26), demonstrating the contribution of Fenton reaction in the observed efficacy. Additionally, the cytotoxicity of **Gi-F-CAA** toward normal cells was further investigated. As obvious in Supplementary Fig. 27, Gi-F exhibited obvious increased cytotoxicity compared with that treated with **Gi-F-CAA**. Subsequently, cellular uptake of **Gi-F-CAA** was performed by confocal laser scanning microscopy (CLSM) (Fig. 4b). As a result, Cy7-labeled **Gi-F-CAA** was barely detected around the EJ cells at pH 7.4. In comparison, obvious fluorescence signal was observed in the cytoplasm at pH 6.5, suggesting that the self-assembled **Gi-F-CAA** possessed a higher internalization capacity by cellular endocytosis. Sub-cellular distribution of **Gi-F-CAA** NPs was investigated by colocalization between **Gi-F-CAA** NPs and lysosomes/mitochondria. As a result, **Gi-F-CAA** were clearly observed inside lysosomes of EJ cells after incubated with EJ cells for 2 h, which demonstrated the endocytosis (Supplementary Fig. 28a). Meanwhile, intensity of red fluorescence signal from NPs increased along with extended incubation time (4 h) whereas it exhibited decreased colocalization with green fluorescence from lysosomes, indicating the lysosomes-escape of NPs. Moreover, NPs were clearly observed to be colocalized with mitochondria at 4 h post-incubation, indicating that lysosomes-escaped **Gi-F-CAA** might get into cytoplasm and locate on mitochondria (Supplementary Fig. 28b). We performed the HPLC assay to investigate the stability of Gi-F in the acidic and enzymatic context of lysosomes. As a result, the peak of Gi-F + Cathepsin (CTS) + lysosomalacidllpase (LAL) (pH 5.0) group was similar to that of Gi-F (pH 7.4) group, indicating the high stability of Gi-F in the acidic and enzymatic context of lysosomes (Supplementary Fig. 29).

To reveal the in vitro self-assembling feature of **Gi-F-CAA**, NBD labeled **Gi-F-CAA** (40 μM) was incubated with EJ cell to observe the NBD fluorescence change by CLSM imaging. Consequently, the EJ cells treated by NBD labeled **Gi-F-CAA** exhibited an increased dotted distributed NBD fluorescence, indicating the self-assembly of **Gi-F-CAA** (Fig. 4c). To further confirm the binding between self-assembled **Gi-F-CAA** and GPX4, EJ cell was treated with NBD labeled **Gi-F-CAA** followed by anti-GPX4 monoclonal antibody treatment. Subsequently, the Pearson correlation coefficient (PCC) was measured with Volocity 3D Image Analysis Software to investigate the co-localization coefficient between **Gi-F-CAA** and GPX4. Once overlapping green fluorescence (**Gi-F-CAA**) with red fluorescence (GPX4) in the confocal image, the high co-localization coefficient of 0.773 clearly verified the efficient interaction between **Gi-F-CAA** and GPX4 (Fig. 4c). In order to demonstrate the ferroptosis mechanism of **Gi-F-CAA** for tumor inhibiting, the intracellular GPX4 activity, GSH and $Fe^{2+}$ ions levels of different treated EJ cells were estimated, respectively. As obvious from Fig. 4d and Supplementary Fig. 30, GPX4 activity in **Gi-F-CAA** (pH 6.5) treated EJ cells was substantially decreased compared with that of other control groups, indicating the enhanced tumor endocytosis and

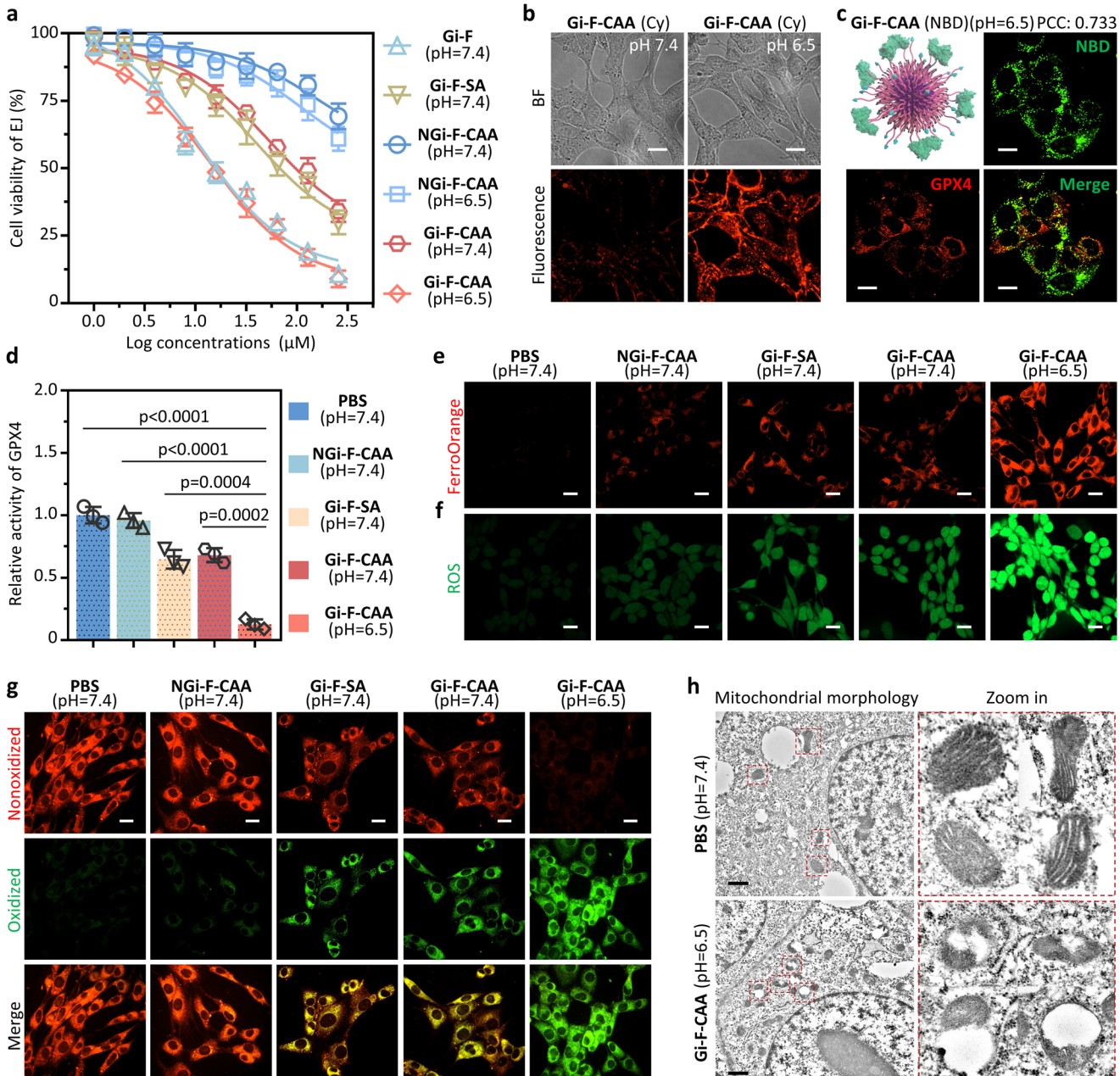

**Fig. 4 | In vivo promoted ferroptosis of Gi-F-CAA by AEB effect. a** Viability of EJ cells after treated with Gi-F (pH = 7.4), Gi-F-SA (pH = 7.4), NGi-F-CAA (pH = 7.4), NGi-F-CAA (pH = 6.5), **Gi-F-CAA** (pH = 7.4) and **Gi-F-CAA** (pH = 6.5) at different concentrations for 48 h (n = 3 experimental repeats). Experiment was independently repeated three times with similar results. **b** Confocal laser scanning microscopy (CLSM) images of EJ cells after treated with **Gi-F-CAA** (pH = 7.4) and **Gi-F-CAA** (pH = 6.5, 40 μM). Scale bar: 10 μm. **c** Fluorescence binding distribution images of NBD labelled **Gi-F-CAA** (pH = 6.5, 40 μM) and GPX4 in EJ cells. Anti-GPX4 antibody was used to label GPX4. Scale bar: 10 μm. **d** Intracellular glutathione peroxidase 4 (GPX4) activities of EJ cells after treated with PBS (pH = 7.4), NGi-F-CAA (pH = 7.4), Gi-F-SA (pH = 7.4), **Gi-F-CAA** (pH = 7.4) and **Gi-F-CAA** (pH = 6.5, 40 μM). The GPX4

activities of PBS group was normalized as 1 (n = 3 experimental repeats). Experiment was independently repeated three times with similar results. **e** Intracellular iron levels of EJ cells after different treatment by FerroOrange fluorescence staining. Scale bar: 10 μm. **f** Intracellular ROS levels of EJ cells after different treatment by DCFH-DA fluorescence staining. Scale bar: 10 μm. **g** Intracellular lipid hydroperoxide (LPO) levels of EJ cells after different treatments by C11-BODIPY staining. Scale bar: 10 μm. **h** Bio-TEM images of EJ cells after treated with PBS (pH = 7.4) and **Gi-F-CAA** (pH = 6.5, 40 μM). The red box dashed region indicates the mitochondria. Scale bar: 10 μm. Data were presented as mean ± SD.
P-values were performed with one-way ANOVA followed by post hoc Tukey's test. Data were expressed as mean ± SD. Source data are provided as a Source Data file.

subsequent **AEB** effect of **Gi-F-CAA** (pH 6.5) by in vivo self-assembly. Moreover, a substantial reduction of GSH levels were detected in EJ cells after **Gi-F-CAA** (pH 6.5) treatment in comparison to that of PBS (pH 7.4), NGi-F-CAA (pH 7.4), Gi-F-SA (pH 7.4) and **Gi-F-CAA** (pH 7.4) treatments (Supplementary Fig. 31). Furthermore, the intracellular generation of $Fe^{2+}$ ions was investigated with a FerroOrange probe, which could produce a bright fluorescent substance with $Fe^{2+}$ ions and

Ferrous Ion Content Assay Kit. As a result, no remarkable fluorescence was observed in EJ cells treated with PBS (pH 7.4), NGi-F-CAA (pH 7.4), Gi-F-SA (pH 7.4) and **Gi-F-CAA** (pH 7.4). By comparison, the EJ cells treated with **Gi-F-CAA** (pH 6.5) emitted a substantially enhanced fluorescence (Fig. 4e), suggesting a vast amount of $Fe^{2+}$ ions generated by assembled **Gi-F-CAA** (Supplementary Fig. 32). Afterwards, the cytoplasmic ROS and LPO levels, as key markers of intracellular

oxidative stress and ferroptosis, were evaluated after treatment with PBS (pH 7.4), NGi-F-CAA (pH 7.4), Gi-F-SA (pH 7.4), **Gi-F-CAA** (pH 7.4) and **Gi-F-CAA** (pH 6.5). Initially, the 2,7-dichlorofluorescein diacetate (DCFH-DA) probe was used to detect intracellular ROS generation. Apparent green fluorescence was observed in **Gi-F-CAA** (pH 6.5) treated EJ cells owing to the Fenton-like reaction-induced oxidative stress (Fig. 4f). Moreover, the **Gi-F-CAA** (pH 6.5) treated EJ cells exhibited an extensive accumulation of LPO compared with that in PBS (pH 7.4), NGi-F-CAA (pH 7.4), Gi-F-SA (pH 7.4) and **Gi-F-CAA** (pH 7.4) treated groups according to the increased green fluorescence and decreased red fluorescence of BODIPY™ 581/591 C11 (Fig. 4g and Supplementary Fig. 33). Afterwards, bio-TEM were further used to evaluate the cellular morphology changes induced by the massive accumulation of LPO. The apparent shrinkage of mitochondrial dimensions, mitochondrial cristae reduction or disappearance and increased membrane density of mitochondria were observed in EJ cells after treated with **Gi-F-CAA** (pH 6.5) compared with that in PBS (pH 7.4) treated groups (Fig. 4h). The results above clearly demonstrated that the **AEB** effect would be beneficial for generating tumor ferroptosis by GPX4 inhibition and ROS generation, resulting in the enhanced anti-tumor capacity.

### In vivo antitumor efficacy of Gi-F-CAA against bladder tumors

Encouraged by the results above, the therapeutic efficacy of **Gi-F-CAA** was evaluated with an established xenografted model of bladder cancer. BALB/c nude mice (~8 weeks, 18 g) were subcutaneously inoculated with $5 \times 10^6$ EJ cells followed by receiving six intravenous injections of PBS, **Gi-F**, Gi-F-SA and **Gi-F-CAA** (8 mg/kg in 100 μL PBS) at 2-days intervals after the average tumor volume reached ~100 mm³ (Fig. 5a). Briefly, the PBS, Gi-F and Gi-F-SA treated mice was used as controls (Fig. 5b, c). A pronounced control of tumor growth was observed in the mice administrated with **Gi-F-CAA** with a mean tumor volume of 552 mm³ compared with that treated with PBS (1910 mm³), Gi-F (1426 mm³) and Gi-F-SA (1204 mm³) at 18 days post injection (Fig. 5c). Meanwhile, all the tumor tissues from different groups were entirely excised for weighting and analysis (Fig. 5d). Accordingly, the tumor growth inhibition (TGI) rate of mice treated with **Gi-F-CAA** was remarkably enhanced to 71%. On the contrary, the mice treated with Gi-F and Gi-F-SA exhibited an inferior TGI for 38% and 47%, respectively (Fig. 5e). Additionally, no noteworthy loss of body weight was detected in mice treated with the **Gi-F-CAA** compared with that treated with PBS (Fig. 5f).

In an attempt to demonstrate the ferroptosis mechanism of **Gi-F-CAA** for tumor inhibiting, the GPX4 activity, iron levels, GSH levels, ROS levels, and LPO levels of different treated tumor tissues were estimated, respectively. As obvious from Fig. 5g, GPX4 activity of tumor tissues in **Gi-F-CAA** group was reduced by 2-times in comparison to that of control groups. Furthermore, the intracellular iron levels for **Gi-F-CAA** treatment tumor tissues were substantially higher than that of other groups, indicating the enhanced endocytosis of **Gi-F-CAA** and subsequent binding with GPX4 by in vivo self-assembly. (Fig. 5h). Cumulatively, a substantial reduction of GSH levels (Fig. 5i) accompanied by the elevation of ROS, PTGS2 and LPO levels (Fig. 5j–l) were detected in tumor tissues following **Gi-F-CAA** treatment compared with that in PBS, Gi-F and Gi-F-SA treatment group. Taken together, these results strongly demonstrated the advantages of **AEB** effect for generating tumor ferroptosis by GPX4 inhibition, thus contributing to a superior tumor control.

### In vivo tumor suppression efficacy of Gi-F-CAA for addressing drug resistance

Accumulating evidences have supported that ferroptosis, as a novel type of regulated cell death, was expected to play a crucial role in developing new therapies for solving tumor drug resistance[31,32]. Subsequently, the therapeutic efficacy of **Gi-F-CAA** was further evaluated

with a multidrug resistance MCF-7/MDR xenografted model. BALB/c nude mice (~8 weeks, 18 g) were subcutaneously inoculated with $5 \times 10^6$ MCF-7/MDR cells followed by receiving six intravenous injections of PBS, DOX (3 mg/kg in 100 μL PBS), CPT (3 mg/kg in 100 μL PBS) and **Gi-F-CAA** (8 mg/kg in 100 μL PBS) every 2 days after the average tumor volume reached ~100 mm³ (Fig. 6a). As consequences, a pronounced control of tumor growth was observed in the mice administrated with **Gi-F-CAA** with a mean tumor volume of 623 mm³ compared with that treated with PBS (1287 mm³), DOX (1082 mm³) and CPT (1095 mm³) at 20 days post injection (Fig. 6b and Supplementary Fig. 34). Meanwhile, all the tumor tissues from different groups were entirely excised for weighting and analysis (Fig. 6c). Accordingly, the TGI rate of mice treated with **Gi-F-CAA** reached to 66%. On the contrary, the mice treated with DOX and CPT exhibited an inferior TGI for 32% and 39%, respectively (Fig. 6d). Additionally, no noteworthy loss of body weight was detected in mice treated with the **Gi-F-CAA** compared with that treated with PBS (Supplementary Fig. 35). Altogether, the results above clearly demonstrated the significantly enhanced ferroptosis-based antitumor efficacy of **Gi-F-CAA** for addressing tumor chemoresistance by **AEB** effect.

### In vivo tumor suppression efficacy of Gi-F-CAA against large tumors

Large tumors were extremely challenging for the application of antibody-based drugs owing to the poor tumor penetrability, resulting in a poor survival of tumor patients in clinics[33,34]. Therefore, the **Gi-F-CAA** were challenged for the treatment of large renal cell carcinoma (~500 mm³) as an inoperable tumor model with a clinical antitumor drug Avastin (Bevacizumab) as control. Subsequently, the BALB/c nude mice (~8 weeks, 18 g) were randomly separated into three groups and subcutaneously inoculated with $5 \times 10^6$ 786-O cells. When tumor volume reached ~500 mm³, the 786-O xenograft mice were administrated with PBS, Bevacizumab (5 mg/kg in 100 μL PBS) and **Gi-F-CAA** (8 mg/kg in 100 μL PBS) at 2-days intervals for a total of six injections (Fig. 6e). Throughout the treatment, the tumor volume in **Gi-F-CAA** group exhibited a declining tendency for 10 days and maintained a long-termed tumor growth inhibition compared with that in the PBS group (Fig. 6f, g), which could be ascribed to the deep tumor penetration of peptide single chains and in vivo self-assembly enhanced anti-tumor capacity. However, owing to the restricted tumor penetration and heterogenous tumor distribution, the antibody squint towards to be localized around tumor vasculature, which limits the efficacy of antibody-based therapies for large tumors. On this basis, the tumor volume of Bevacizumab group exhibited a temporary retarded growth and got out of control thereafter (Fig. 6f, g). These results altogether demonstrated the substantially enhanced ferroptosis-based antitumor efficacy of **Gi-F-CAA** for resolving large tumors in clinic, which might be attributed to the promoted drug delivery efficacy by deep tumor penetration, improved endocytosis, and **AEB** effect in cells.

### Biosafety and toxicity profile evaluation of Gi-F-CAA

Lastly, to further evaluate the potential in vivo toxicology of **Gi-F-CAA**, the main organs and serum of healthy BALB/c nude mice (~8 weeks, 18 g) were harvested after administrated with PBS, **Gi-F-CAA** (8 mg/kg in 100 μL PBS), **Gi-F-CAA** (12 mg/kg in 100 μL PBS) and **Gi-F-CAA** (16 mg/kg in 100 μL PBS) by intravenous injection. As obvious in Supplementary Fig. 36, no remarkable histopathological damages were observed according to the hematoxylin and eosin (H&E) images of major organs (including heart, liver, spleen, lungs and kidneys) at 1 day post-injection. Meantime, hepatic and renal function biochemical markers tests exhibited normal ranges of values in the PBS, **Gi-F-CAA** (8 mg/kg in 100 μL PBS), **Gi-F-CAA** (12 mg/kg in 100 μL PBS) and **Gi-F-CAA** (16 mg/kg in 100 μL PBS) treatment groups at 1 day post-injection, including alanine aminotransferase (ALT), aspartate aminotransferase

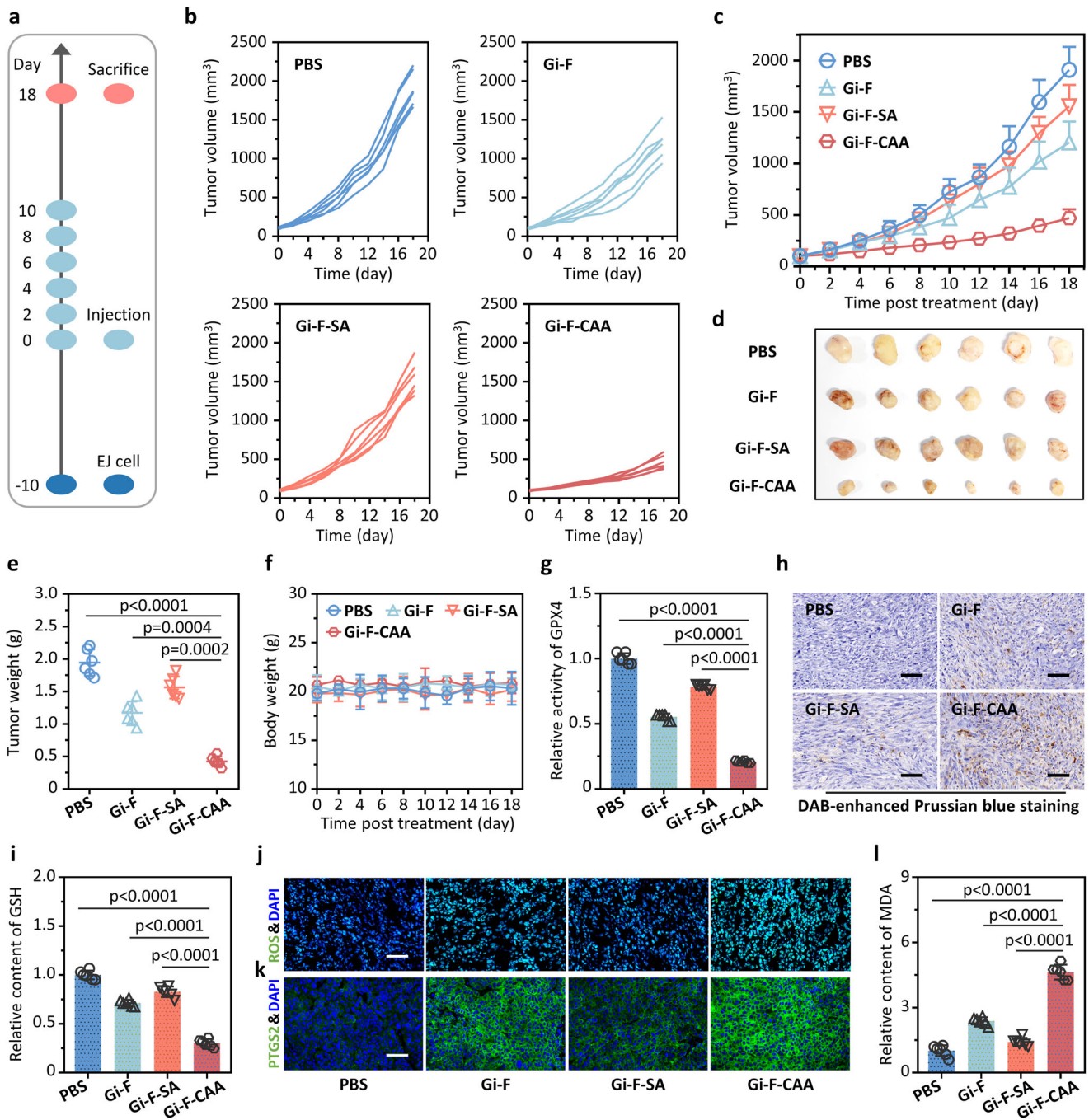

**Fig. 5 | Antitumor efficacy of Gi-F-CAA against EJ xenograft tumors. a** BALB/c nude mice were subcutaneously inoculated with $5 \times 10^6$ EJ cells and intravenously administrated with PBS, Gi-F (8 mg/kg in 100 μL PBS), Gi-F-SA (8 mg/kg in 100 μL PBS) and **Gi-F-CAA** (8 mg/kg in 100 μL PBS). **b** The individual tumor growth curves of mice in different treatment groups. **c** The average tumor growth curves of EJ xenografted mice after different treatments over 18 days ($n$ = 6 mice). **d** EJ tumor tissues were harvested at 18 days after different treatments. **e** EJ tumor weights after different treatments over 18 days ($n$ = 6 mice). **f** Body weight changes of EJ xeno-grafted mice after different treatments over 18 days ($n$ = 6 mice). **g** GPX4 activities of EJ tumor tissues after different treatments ($n$ = 6 mice). The GPX4 activities of PBS group was normalized as 1. **h** Iron levels in EJ tumor tissues after different treatments by DAB-enhanced Prussian blue iron staining. Scale bar: 50 μm. **i** Glutathione (GSH) levels of EJ tumor tissues after different treatments ($n$ = 6 mice). The GSH levels of PBS group was normalized as 1. **j** ROS levels of EJ tumor tissues after different treatments by DCFH-DA fluorescence staining. Scale bar: 50 μm. **k** Immunofluorescence analyses of Prostaglandin-Endoperoxide Synthase 2 (PTGS2) expression in EJ tumor tissues after different treatments. Scale bar: 50 μm. **l** Malonaldehyde (MDA) levels of EJ tumor tissues after different treatments ($n$ = 6 mice). The MDA levels of PBS group was normalized as 1. *P*-values were performed with one-way ANOVA followed by post hoc Tukey's test. Data were expressed as mean ± SD. Source data are provided as a Source Data file.

(AST), alkaline phosphatase (ALP), urea nitrogen (BUN), and creatinine (CRE) levels (Supplementary Fig. 37). Additionally, the results of blood routine examination including white blood cell (WBC), red blood cell (RBC), platelets (PLT), lymphocyte (LYMPH) and neutrophil (NEUT) after treatment with **Gi-F-CAA** (8 mg/kg in 100 μL PBS), **Gi-F-CAA** (12 mg/kg in 100 μL PBS) and **Gi-F-CAA** (16 mg/kg in 100 μL PBS) for 1 day were comparable with that of PBS group (Supplementary Fig. 38). All the above results demonstrated the excellent biocompatibility profile of **Gi-F-CAA** in clinical translational applications of biomedical areas.

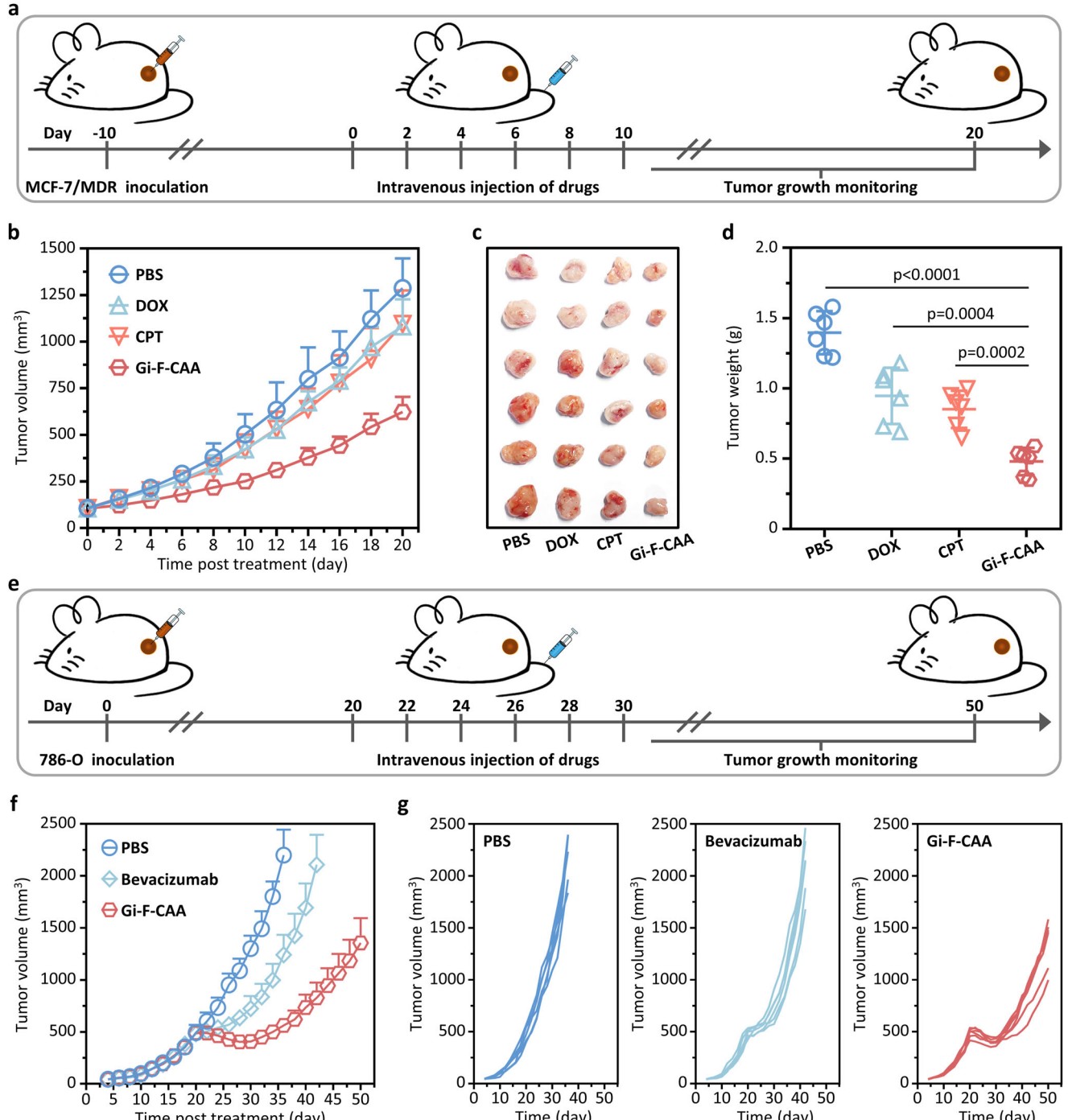

**Fig. 6 | In vivo tumor suppression efficacy of Gi-F-CAA for addressing drug resistance. a** BALB/c nude mice were subcutaneously inoculated with $5 \times 10^6$ MCF-7/MDR cells and intravenously administrated with PBS, DOX (3 mg/kg in 100 µL PBS), CPT (3 mg/kg in 100 µL PBS) and **Gi-F-CAA** (8 mg/kg in 100 µL PBS). **b** The average tumor growth curves of MCF-7/MDR xenografted mice after treated with PBS, DOX, CPT and **Gi-F-CAA** over 20 days ($n = 6$ mice). **c** MCF-7/MDR tumor tissues were harvested at 20 days post-injection. **d** MCF-7/MDR tumor weights after treated with PBS, DOX, CPT and **Gi-F-CAA** over 20 days ($n = 6$ mice). **e** BALB/c nude mice were subcutaneously inoculated with $5 \times 10^6$ 786-O cells and intravenously administrated with PBS, Bevacizumab (5 mg/kg in 100 µL PBS) and **Gi-F-CAA** (8 mg/ kg in 100 µL PBS). **f** The average tumor growth curves of 786-O xenografted mice after treated with PBS, Bevacizumab and **Gi-F-CAA** ($n = 6$ mice). **g** The individual tumor growth curves of mice in the PBS group, Bevacizumab group and **Gi-F-CAA** group. *P*-values were performed with one-way ANOVA followed by post hoc Tukey's test. Data were expressed as mean ± SD. Source data are provided as a Source Data file.

## Discussion

In terms of biomedical applications of nanomaterials, many pioneering works about nanoparticle-induced ferroptosis have been reported, especially with regard to the exploration of nano-structure-activity relationships (nano-SARs), which realized significant tumor inhibition effects[35–37]. In this work, we explored the biological effect of nano-SARs, by developing a microenvironment-responsive in vivo self-assembled peptide-fer-riporphyrin conjugates (**Gi-F-CAA**) for improving tumor penetration, accumulation and binding affinity with GPX4, ultimately synergistically enhancing ferroptosis-based anticancer activity. The in vivo self-assembly strategy based on supramolecular

chemistry substantially enhanced the accumulation of functional agents to its action sites by overcoming the physical or metaphorical barriers. On this basis, the designed **Gi-F-CAA** with assembly enhanced binding (AEB) effect demonstrated substantial inhibition of GPX4 activity by the formation of self-assembled Gi-F nanoparticles, which augmented the oxidative stress of Fenton reaction induced by FeTCPP, leading to remarkably enhanced tumor ferroptosis. According to the in vitro *and* in vivo experiment results, the in vivo self-assembly and multivalent cooperative interactions could be intelligently and precisely realized, resulting in the amazing ferroptosis-based antitumor properties in multiple tumor models, which provides an alternative strategy for malignant tumor therapy. Although the understanding of ferroptosis mechanisms and implications has significantly progressed, there are still no drugs specifically targeting ferroptosis for clinical use currently. Despite the limitations, ferroptosis research shows promise for the development of innovative treatments for various diseases. By combination with other therapies, ferroptosis has the potential to improve their efficacy and revolutionize cancer treatment.

## Methods

### Ethical statement

All the animal experiments were conducted in compliance with ethical compliance and approval by the Institutional Animal Care and Use Committee of National Center for Nanoscience and Technology (NCNST21-2208-0608). All the experiments performed with human specimens were reviewed and approved by the Committees for Ethical Review of the Fourth Hospital of Harbin Medical University (2022-SCILLSC-28). Written informed consent was obtained from bladder cancer patients.

### The use of bladder cancer patient samples

Bladder cancer patients were randomly recruited to collect bladder tumor tissues and normal bladder tissues. Inclusion criteria: (1) Bladder cancer patients who have signed the informed consent form before any study-related procedures. Patients have fully understood the study and voluntarily signed the informed consent form in writing. (2) Intended for surgical treatment. (3) Aged between 18 and 70. (4) Karnofsky Performance Status (KPS) score $\geq 60$. (5) Basic normal organ function; neutrophil count $> 1.5 \times 109/L$, platelet count $> 100 \times 109/L$, hemoglobin $> 9.0$ g/dL. Total bilirubin is normal or $< 1.5 \times$ ULN; AST (SGOT) and ALT (SGPT) $< 2.5 \times$ ULN (if liver metastasis is present $< 5 \times$ ULN); serum creatinine is $< 1.5 \times$ ULN. (6) Expected survival period $\geq 3$ months. Exclusion criteria: (1) Patients with abnormalities in the urinary system or urethral stricture. (2) Pregnant or lactating women. (3) Patients with severe concomitant diseases in other systems. (4) Patients allergic to multiple drugs. (5) Patients who are currently participating or have participated in another clinical study within 30 days. None of the above inclusion criteria or exclusion criteria affected the experimental results. Corresponding patients were treated as standard of care. Sex and gender of bladder cancer patient has been collected and recorded by authors, which has no bearing on data analysis or results.

### Materials

Peptides were purchased from Nanjing Peptide Biotech LTD. (Nanjing, China). Glutathione (GSH) Assay Kit (BC1175), Cell Iron Content Assay Kit (BC5310), Reactive Oxygen Species (ROS) Assay Kit (CA1410), Glutathione Peroxidase (GPX) Assay Kit (BC1195) and Malondialdehyde (MDA) Assay Kit (BC0025) were purchased from Beijing Solarbio Science & Technology Co., Ltd. (Beijing, China). LysoTracker Green (C1047S) and Mito-Tracker Green (C1048) were purchased from beyotime biotechnology co. ltd. Penicillin and streptomycin (CK0008-100ML), CCK-8 Cell Proliferation and Cytotoxicity Assay Kit (CK0001-100T) and 4% paraformaldehyde (CK0014-500ML) were purchased from Beijing Chengzhi Kewei Biotechnology Co. Ltd. GPX4 Protein (29993-H07E) was purchased from Sino Biological Inc. Anti-Glutathione Peroxidase 4 (ab125066) was purchased from Abcam Trading Co. Ltd. 96-Well Plates (MBB-96-LD) and Mycoplasma Removal Agent (MRA, M1056) were purchased from LABLEAD Inc. (Beijing, China). EJ bladder cancer cell line, MCF-7/MDR breast cancer cell line and 786-O renal cell carcinoma cell line were purchased from the Cell Culture Center of the Institute of Basic Medicine, Chinese Academy of Medical Sciences (Beijing, China). The cell lines were not tested for mycoplasma contamination and authenticated. Female BALB/c nude mice (8 weeks old, 18 g) were purchased from Vital River Laboratory Animal Technology Co., Ltd. (Beijing, China).

### Animals and cells

For the in vivo tumour experiments, the maximum permitted tumour burden occurred when the tumour volume exceeded 2000 mm³. Female BALB/c nude mice (~8 weeks, 18 g) were purchased from Vital River Laboratory Animal Technology Co., Ltd. (Beijing, China). Mice were housed in a ventilated cage ($n = 5$) with 12 h light-dark cycle, constant room temperature at 22 °C, a relative humidity of 30–70% and food, water ad libitum. EJ bladder cancer cell lines, MCF-7/MDR breast cancer cell lines and 786-O renal cell carcinoma cell lines were purchased from the Cell Culture Center of the Institute of Basic Medicine, Chinese Academy of Medical Sciences (Beijing, China), which were cultured in Dulbecco's Modified Eagle Medium (DMEM) or Roswell Park Memorial Institute (RPMI) 1640 Medium containing 10% fetal bovine serum, 1% penicillin and streptomycin, 0.1% Mycoplasma Removal Agent at 37 °C in 5% $CO_2$.

### Synthesis and characterization of Gi-F-CAA

Experiments used peptide TCPP-LKLKLKGACNWLPLYPCPV and control peptide TCPP-LKLKLKGACLNWPYLPCPV were purchased from Nanjing Peptide Biotech LTD. The molecular weight and purity were characterized by matrix-assisted laser desorption/ionization time-of-flight mass spectrometry (MALDI-TOF-MS) and high-performance liquid chromatography (HPLC). Then the peptide TCPP-LKLKLKGACNWLPLYPCPV (20 mg, 0.007 mmol) was dissolved in 2 mL of 0.1 M $NaHCO_3$ solution (pH 9), followed by adding 5 mg CAA dissolved in 2 mL of $NaHCO_3$ solution under stirring. After reaction for overnight at room temperature, the mixture was dialyzed against deionized water (MWCO: 1000 Da) for 24 h and lyophilized. The $FeCl_3$ was added into lyophilized powder and dispersed in DMSO for 12 h, and the final product of **Gi-F-CAA** was obtained after dialysis. The control peptide-porphyrin conjugates, including Gi-F-SA (FeTCPP-LKLKLK(SA)GACNWLPLYPCPV, conjugates decorated with acid-nonsensitivity succinic anhydride moieties), NGi-F-CAA (FeTCPP- LKLKLK(CAA)GACLNWPYLPCPV, conjugates without GPX4 inhibition ability), and Gi-F (FeTCPP-LKLKLKGACNWLPLYPCPV, conjugates with nanoparticle forming ability) were prepared by the same method.

### Particle size and morphology characterization

Nano-ZS 3600 (Malvern Instruments, UK) was used to measure the size distribution of the peptide-porphyrin conjugate solutions. The conjugate solutions (40 μM) were incubated at pH 7.4 or pH 6.5 before measurements. An average value was determined by three repeated measurements at 25 °C for each sample. Besides, transmission electron microscopy (TEM, Tecnai G2 20 S-TWIN) was conducted to observe the morphology and diameters of conjugates in aqueous solution, The conjugates (50 μL, 40 μM) were kept in PB solution at pH 7.4 or pH 6.5, which were dropped on copper meshes. After 5 min, unnecessary liquid was removed and the uranyl acetate solution (10 μL) was dropped on this copper mesh to stain the samples for 10 min. Finally, the

copper meshes were washed with deionized water, and dried at room temperature for observation.

## OH detection by EPR spectrometer and TPA fluorescence spectra assay

The generation of •OH was determined by the Magnettech MS-5000 EPR spectrometer. Typically, peptide-porphyrin conjugates (40 μM) were exposed to $H_2O_2$ (200 μM) for 3 min in the presence of 5,5-dimethyl-1-pyrroline-N-oxide (DMPO, 100 μM). The •OH signal was immediately detected by the EPR spectrometer. The amount of •OH was measured by using TPA as an indicator that following the increasement of HTPA fluorescence in the conjugate solution. Briefly, TPA (50 μL, 2 mM) and fresh conjugate solution (50 mM) were mixed with 1 ml $H_2O$, and then the solution was exposed to various concentration of $H_2O_2$. The fluorescence at 425 nm (λex = 310 nm) was recorded at 5 min to determine the concentration of •OH, which can further be applied for calculating the TOF of peptide-porphyrin conjugates.

## Critical aggregation concentration (CAC)

The CACs were determined using pyrene as a fluorescent probe. In brief, 50 μL of pyrene solution in acetone (480 μM) was added to a 5 mL centrifuge tube, followed by adding a series of peptide-porphyrin conjugate solutions with different concentrations, and the tube was kept overnight to evaporate the acetone completely, the concentration of pyrene being fixed at 6 μM at last. The excitation spectra of the solutions were recorded with the emission wavelength fixing at 393 nm, and the intensity ratio ($I_{338}/I_{335}$) was calculated as a function of the concentration. The CAC was determined based on the intersection point at low concentration on the plot.

## Hydrolysis analysis of the peptide-porphyrin conjugates

In vitro hydrolysis analysis of the peptide-porphyrin conjugates was studied in pH values of 7.4 and 6.5 solutions. 20 mg of peptide-porphyrin conjugate was dissolved in 2 mL pH 7.4 buffer solution, which was then transferred into a dialysis bag (MWCO: 1000 Da) and immersed in 25 mL of different pH buffer solutions at 37 °C with continuous shaking respectively. At fixed time intervals (20 min, 40 min, 1 h, 2 h and 4 h), 1 mL of dialysate from the outer fluid was collected analyze the hydrolyzed CAA concentration for each group by a high-performance liquid chromatography (HPLC, WATERS Alliance HPLC e2489), and an equal volume of fresh buffer solution was replaced to the system. In order to determine the CAA concentration according to the peak area of the CAA, a standard curve of the CAA was plotted, with the peak area as a function of the concentration. The hydrolysis degree was then calculated through quantifying hydrolyzed CAA concentration at each time.

## Computational structure preparations

The structure of the porphyrin part in the peptide was constructed in Molecular Operating Environment (MOE2020) software and optimized under B3LYP method with Lanl2DZ basis set for Fe atom and 6−31 G* basis set for other atoms in Gaussian16 software. Meanwhile, the decorated lysine (deLys) residue was generated based on the lysine residue and also optimized under the same method and basis set (B3LYP/6−31 G*) in Gaussian16 software. The charges of the porphyrin part and the deLys residue were calculated by the restricted electrostatic potential (RESP)[38]. And their geometry properties (i.e., force field parameter) of the two customize residues were generated using antechamber package in AmberTools based on the General Amber Force Field (GAFF)[39]. After building the parameters of the porphyrin and the deLys, both the Gi-F-SA and the Gi-F were constructed by the leap package in Amber 20[40].

## Molecular dynamics simulations

Since both Gi-F-SA and Gi-F were built from the sequence, the relaxation of the structures was necessary for the further simulations. Therefore, at firstly, the structure optimizations and molecular dynamics (MD) simulations for the single peptides were employed to obtain the relaxation structure of the Gi-F-SA and the Gi-F. Based on the relaxation structure of the Gi-F-SA, the Packmol[41] software was applied to put 24 Gi-F-SAs randomly into a 100 Å *100 Å *100 Å box and sufficient numbers of water molecules were put into the same box to fill the empty space. And based on the relaxation structure of the Gi-F, the Packmol software[41] was also applied to put 24 Gi-Fs into a 100 Å *100 Å *100 Å box but they were not randomly distribution but formed a nanosphere in which the porphyrin parts were distributed in the inside and other parts in the outside. And also, sufficient numbers of water molecules were put into the same box to fill the empty space.

Both the randomly distributed Gi-F-SAs system and Gi-Fs nanosphere system were finally setup by tleap package in Amber20, and their molecular dynamics simulations were run by the pmemd.cuda package in Amber20[40]. The simulations went through multiple steps. Firstly, each system was minimized with the steepest descend method and conjugated gradient method. After that, the Langevin thermostat[41] was applied to heat up the system to 300 K. And the NPT ensemble was employed to equilibrate the system to the 1 atmosphere pressure. Finally, the long-time MD simulations (200 ns) were run under NVT ensemble. During the MD simulations, except the customize porphyrin residue and deLys residue, the other residues and water molecules in the systems were describe by the FF19SB force field[42] and TIP3P model[43]. And to maintain the coordination of the Fe ion, several constrains were set on the distances of the Fe ion and its coordinated atoms. Meanwhile, the Shake algorithm[44] was applied to constrain the bonded hydrogen movement in the MD simulations.

## Microscale Thermophoresis (MST) ligand binding measurements

Initially, GPX4 was labeled using Protein Labeling Kit RED-NHS according to the manufacturer's instructions to investigate the binding affinity of **Gi-F-CAA**, Gi-F and NGi-F-CAA with GPX4 protein. Afterward, the labelled GPX4 protein was incubated with series of **Gi-F-CAA**, Gi-F and NGi-F-CAA concentration gradients. Finally, the prepared samples were loaded into silica capillaries and measured with Monolish NT.115. Apparent Kd represented the affinity of assemblies rather than a single molecule. Bovine serum albumin (BSA) was used as control for calculating the apparent binding affinity between Gi-F (nanoparticle) to GPX4 protein. Briefly, as the diameter of nanoparticle is proportional to the molecular weight, according to the reported bovine serum albumin (BSA) protein size and molecular weight (7.2 nm, 66.43 KDa), the molecular weight of Gi-F nanoparticle with an average 78 nm size can be calculated to be 719.66 KDa. The monomer amount Gi-F nanoparticle was calculated as 84.46/2.927 = 246.

## In vivo and ex vivo fluorescence imaging of Gi-F-CAA

Firstly, A density of $5 \times 10^6$ EJ cell resuspended in Matrigel were subcutaneously injected into the right hind of mice to construct EJ xenograft model. When the tumor volume reached about 200–400 mm³, the mice were administrated with Cy labeled Gi-F (8 mg/kg in 100 μL PBS), Cy labeled Gi-F-SA (8 mg/kg in 100 μL PBS) and Cy labeled **Gi-F-CAA** (8 mg/kg in 100 μL PBS) by intravenous injection for in vivo fluorescence imaging at different timepoints (10 min, 20 min, 40 min, 1 h, 2 h, 4 h, 8 h, 12 h and 24 h) with in vivo imaging system (IVIS). Meanwhile, the EJ xenograft mice were sacrificed for collecting major organs and tumors to evaluate the biodistribution of **Gi-F-CAA**.

## Penetration assay in tumor frozen sections

Firstly, EJ xenograft mice were administrated with Cy labeled Gi-F (8 mg/kg in 100 μL PBS) and Cy labeled **Gi-F-CAA** (8 mg/kg in 100 μL

PBS) by intravenous injection. Afterward, EJ tumor tissues were entirely resected and protected from light. Subsequently, the tumor tissues were fixed with optimum cutting temperature (OCT) compound for 12 h at −20 °C avoiding light. Finally, the fixed tumor tissues were cut into sections and immediately observed with UltraVIEW VoX.

## Frozen sections of tumor tissue

Firstly, EJ xenograft mice were administrated with NBD labeled Gi-F-SA (8 mg/kg in 100 μL PBS) and NBD labeled **Gi-F-CAA** (8 mg/kg in 100 μL PBS) by intravenous injection. Afterward, EJ tumor tissues were resected and protected from light. Subsequently, the tumor tissues were fixed with optimum cutting temperature (OCT) compound for 12 h at −20 °C avoiding light. Finally, the fixed tumor tissues were cut into sections and immediately observed with UltraVIEW VoX.

## Cellular imaging experiment of Gi-F-CAA

Initially, EJ cells ($5 \times 10^5$) were cultured in confocal dish with a $CO_2$ incubator overnight at 37 °C. Subsequently, the EJ cells were incubated with Cy labeled **Gi-F-CAA** (40 μM, pH = 7.4) and Cy labeled **Gi-F-CAA** (40 μM, pH = 6.5) for 2 h followed by washing with PBS for three times. Finally, the fluorescence was observed with Confocal Laser Scanning Microscope (CLSM) under a 40 × objective (Ex/Em: 640 nm/725 nm).

## Colocalization assay by confocal laser scanning microscopy (CLSM)

Initially, EJ cells ($5 \times 10^5$) were cultured in confocal dish with a $CO_2$ incubator overnight at 37 °C. Subsequently, the EJ cells were incubated with Cy labeled **Gi-F-CAA** (40 μM, pH = 6.5) for 2 h or 4 h followed by washing with PBS for three times. Afterward, the lysosomes of cells were stained with LysoTracker Green and the mitochondria of cells were stained with MitoTracker Green at 37 °C for 30 min. Finally, the fluorescence was observed with Confocal Laser Scanning Microscope (CLSM) under a 40 × objective.

## Cytotoxicity assay of Gi-F-CAA

Initially, EJ cells were seeded into 96 well plates at a density of $5 \times 10^3$ per well and incubated in a $CO_2$ incubator overnight at 37 °C. Subsequently, Gi-F (pH = 7.4), Gi-F-SA (pH = 7.4), NGi-F-CAA (pH = 7.4), NGi-F-CAA (pH = 6.5), **Gi-F-CAA** (pH = 7.4) and **Gi-F-CAA** (pH = 6.5) at different concentrations (1, 2, 4, 8, 16, 32, 64, 128 and 256 μM) was added to the cells and further incubated for 48 h. Finally, the cell viability rate was calculated by (Sample wells-Blank wells)/(Control wells-Blank wells) × 100%.

## Intracellular GSH content detection

Initially, EJ cells ($5 \times 10^5$) were cultured in 6-well plate with a $CO_2$ incubator overnight at 37 °C. Subsequently, EJ cells were exposed to different culture mediums containing PBS (pH = 7.4), NGi-F-CAA (pH = 7.4), Gi-F-SA (pH = 7.4), **Gi-F-CAA** (pH = 7.4) and **Gi-F-CAA** (pH = 6.5, 40 μM) for another 8 h, respectively. Next, EJ cells were collected and washed with PBS for 3 times. Afterwards, the cell lysates were collected by centrifugation at 8000 g for 10 min at 4 °C after lysed by freeze-thaw treatment. Finally, the GSH contents were measured with a GSH Assay Kit based on the manufacturer's instructions.

## Intracellular $Fe^{2+}$ ions detection

Initially, EJ cells ($5 \times 10^5$) were cultured in confocal dish with a $CO_2$ incubator overnight at 37 °C. Subsequently, EJ cells were exposed to different culture mediums containing PBS (pH = 7.4), NGi-F-CAA (pH = 7.4), Gi-F-SA (pH = 7.4), **Gi-F-CAA** (pH = 7.4) and **Gi-F-CAA** (pH = 6.5, 40 μM) for another 8 h, respectively. Afterwards, EJ cells were washed with PBS for 3 times and replaced with culture mediums. Finally, the intracellular iron levels in EJ cells were detected with a FerroOrange fluorescence staining based on the manufacturer's

instructions, which was observed with Confocal Laser Scanning Microscope (CLSM) under a 40 × objective (Ex/Em: 488 nm/525 nm).

## Intracellular ROS detection

Initially, EJ cells ($5 \times 10^5$) were cultured in confocal dish with a $CO_2$ incubator overnight at 37 °C. Subsequently, EJ cells were exposed to different culture mediums containing PBS (pH = 7.4), NGi-F-CAA (pH = 7.4), Gi-F-SA (pH = 7.4), **Gi-F-CAA** (pH = 7.4) and **Gi-F-CAA** (pH = 6.5, 40 μM) for another 8 h, respectively. Afterwards, EJ cells were washed with PBS for 3 times and replaced with culture mediums. Finally, the production of ROS in EJ cells was detected with a ROS Assay Kit based on the manufacturer's instructions, which was observed with Confocal Laser Scanning Microscope (CLSM) under a 40 × objective (Ex/Em: 488 nm/525 nm).

## Intracellular LPO fluorescence imaging

Initially, EJ cells ($5 \times 10^5$) were cultured in confocal dish with a $CO_2$ incubator overnight at 37 °C. Subsequently, EJ cells were exposed to different culture mediums containing PBS (pH = 7.4), NGi-F-CAA (pH = 7.4), Gi-F-SA (pH = 7.4), **Gi-F-CAA** (pH = 7.4) and **Gi-F-CAA** (pH = 6.5, 40 μM) for another 8 h, respectively. Next, EJ cells were collected and washed with PBS for 3 times. Finally, the production of LPO in EJ cells were detected with BODIPY™ 581/591 C11 based on the manufacturer's instructions, which was observed with Confocal Laser Scanning Microscope (CLSM) under a 40 × objective (Ex/Em: 488 nm/525 nm).

## Intracellular LPO content detection

Initially, EJ cells ($5 \times 10^5$) were cultured in 6-well plate with a $CO_2$ incubator overnight at 37 °C. Subsequently, EJ cells were exposed to different culture mediums containing PBS (pH = 7.4), NGi-F-CAA (pH = 7.4), Gi-F-SA (pH = 7.4), **Gi-F-CAA** (pH = 7.4) and **Gi-F-CAA** (pH = 6.5, 40 μM) for another 8 h, respectively. Next, EJ cells were collected and washed with PBS for 3 times. Afterwards, the cell lysates were collected by centrifugation at 8000 g for 10 min at 4 °C after lysed by freeze-thaw treatment. Finally, the LPO contents were measured by thiobarbituric acid (TBA) reaction with a Malondialdehyde (MDA) Assay Kit based on the manufacturer's instructions.

## Intracellular GPX4 activity detection

Initially, EJ cells ($5 \times 10^5$) were cultured in 6-well plate with a $CO_2$ incubator overnight at 37 °C. Subsequently, EJ cells were exposed to different culture mediums containing PBS (pH = 7.4), NGi-F-CAA (pH = 7.4), Gi-F-SA (pH = 7.4), **Gi-F-CAA** (pH = 7.4) and **Gi-F-CAA** (pH = 6.5, 40 μM) for another 8 h, respectively. Next, EJ cells were collected and washed with PBS for 3 times. Afterwards, the cell lysates were collected by centrifugation at 5000 rpm for 10 min at 4 °C after ultrasonic lysis in an ice bath. Finally, the GPX4 activity were measured with a GPX Assay Kit based on the manufacturer's instructions.

## Bio-TEM imaging of EJ cells

Initially, EJ cells ($5 \times 10^5$) were cultured in 6-well plate with a $CO_2$ incubator overnight at 37 °C. Subsequently, EJ cells were exposed to different culture mediums containing PBS (pH = 7.4) and **Gi-F-CAA** (pH = 6.5, 40 μM) for another 8 h, respectively. Next, EJ cells were collected and fixed in 2.5% glutaraldehyde overnight at 4 °C. Afterwards, the EJ cells were dehydrated with acetone solutions, which were infiltrated with Epon 812 at 4 °C overnight followed by solidifying. Finally, the samples were sectioned, stained and characterized through HT7700 Transmission Electron Microscope.

## In vivo antitumor evaluation of Gi-F-CAA in EJ xenograft model

Female BALB/c nude mice (~8 weeks, 18 g) were subcutaneously inoculated with $5 \times 10^6$ EJ cells resuspended in Matrigel. When the

tumor size reached about 100 mm³, EJ tumor-bearing mice were randomly divided into four groups and administered with PBS, Gi-F (8 mg/kg in 100 μL PBS), Gi-F-SA (8 mg/kg in 100 μL PBS) and **Gi-F-CAA** (8 mg/kg in 100 μL PBS) by intravenous injection at 2-days intervals for six times. Individual tumor volume and bodyweight of mouse after different treatment was collected at 2-days intervals for analysis. Mice were killed at 18 days post treatment unless endpoints were exceeded. Moreover, tumors from all groups were harvested under euthanasia and weighed for analysis at the end of treatment.

### Tumor iron content detection
EJ tumor tissues were collected under euthanasia at the end of treatment and washed with PBS for 3 times. Afterwards, the DAB-enhanced Prussian blue iron staining was performed with Wuhan Servicebio Technology Co., Ltd.

### Tumor PTGS2 content detection
EJ tumor tissues were collected under euthanasia at the end of treatment and washed with PBS for 3 times. Afterwards, the immunofluorescence analyses of Prostaglandin-Endoperoxide Synthase 2 (PTGS2) expression were performed with Wuhan Servicebio Technology Co., Ltd.

### Tumor GPX4 activity detection
EJ tumor tissues were collected under euthanasia at the end of treatment and washed with PBS for 3 times. Afterwards, the tumor lysates were collected by centrifugation at 5000 rpm for 10 min at 4 °C after lysed by freeze-thaw treatment. Finally, the GPX4 activity were measured with a GPX Assay Kit based on the manufacturer's instructions.

### Tumor GSH content detection
EJ tumor tissues were collected under euthanasia at the end of treatment and washed with PBS for 3 times. Afterwards, the tumor lysates were collected by centrifugation at 8000 rpm for 10 min at 4 °C after lysed with Frozen Type Tissuelyser. Finally, the GSH contents were measured with a GSH Assay Kit based on the manufacturer's instructions.

### Tumor ROS content detection
EJ tumor tissues were collected under euthanasia at the end of treatment and washed with PBS for 3 times. Next, DCFH-DA fluorescence staining was carried out by Wuhan Servicebio Technology Co., Ltd.

### Tumor MDA content detection
EJ tumor tissues were collected under euthanasia at the end of treatment and washed with PBS for 3 times. Afterwards, the tumor lysates were collected by centrifugation at 8000 g for 10 min at 4 °C after lysed with Frozen Type Tissuelyser. Finally, the MDA contents were measured by thiobarbituric acid (TBA) reaction with a Malondialdehyde (MDA) Assay Kit (Solarbio Science & Technology Co., Ltd., China, BC0025) based on the manufacturer's instructions.

### In vivo antitumor evaluation of Gi-F-CAA in MCF-7/MDR xenografted model
Female BALB/c nude mice (~8 weeks, 18 g) were subcutaneously inoculated with $5 \times 10^6$ MCF-7/MDR cells resuspended in Matrigel. When the tumor size reached about 100 mm³, MCF-7/MDR xenografted mice were randomly divided into four groups and administrated with PBS, DOX (3 mg/kg in 100 μL PBS), CPT (3 mg/kg in 100 μL PBS) and **Gi-F-CAA** (8 mg/kg in 100 μL PBS) by intravenous injection at 2-days intervals for six times. Individual tumor volume and bodyweight of mouse after different treatment was collected at 2-days intervals for analysis. Mice were killed at 20 days post treatment unless endpoints were exceeded. Moreover, tumors from all groups were harvested under euthanasia and weighed for analysis at the end of treatment.

### In vivo antitumor evaluation of Gi-F-CAA in large tumor model
Female BALB/c nude mice (~8 weeks, 18 g) were subcutaneously inoculated with $5 \times 10^6$ 786-O cells resuspended in Matrigel and randomly divided into three groups. When the tumor size reached about 500 mm³, 786-O xenografted mice were administrated with PBS, Bevacizumab (5 mg/kg in 100 μL PBS) and **Gi-F-CAA** (8 mg/kg in 100 μL PBS) by intravenous injection at 2-days intervals for six times. Individual tumor volume and bodyweight of mouse after different treatment was collected at 2-days intervals for analysis.

### In vivo toxicology evaluation of Gi-F-CAA
The main organs (heart, liver, spleen, lung, kidney and intestine) of healthly BALB/c nude mice (~8 weeks old, 18 g) were collected under euthanasia after treatment with PBS and **Gi-F-CAA** (8, 12 and 16 mg/kg in 100 μL PBS) for 1 day. Afterwards, the histology evaluation by Hematoxylin and Eosin (H&E) Staining was carried out by Servicebio Technology Co., Ltd. (Wuhan, China). Meanwhile, the whole blood and serum of healthly BALB/c nude mice (~8 weeks old, 18 g) were collected after treatment with PBS and **Gi-F-CAA** (8, 12 and 16 mg/kg in 100 μL PBS) for 1 days. The blood biochemistry analysis was performed with Hitachi Automatic Biochemical Analyzer 7100. The blood routine analysis was carried out by Vital River Laboratory Animal Technology Co., Ltd. (Beijing, China).

### Statistical methods
Data analysis was carried out with one-way ANOVA followed by post hoc Tukey's test. NS means no significance, $P < 0.05$ was considered statistically significant. Data were presented as mean ± SD.

### Reporting summary
Further information on research design is available in the Nature Portfolio Reporting Summary linked to this article.

## Data availability
The data supporting the findings of this study are available within the article and Supplementary Information files. Source data are provided with this paper. Correspondence and requests for materials should be addressed to T.-L.S., Z.-Y.Q., W.-H.X. and H.W. (wanghao@nanoctr.cn).

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

## Acknowledgements

This work was supported by the National Key R&D Program of China (2021YFB3801000), Regional Key Project of National Natural Science Foundation of China (U20A20385), National Natural Science Foundation of China (22222503), the Strategic Priority Research Program of the Chinese Academy of Sciences (XDB36000000) and Overseas High-end Talents Introduction Program (G2022011023L). Z.-Y.Q. thanks the Youth Innovation Promotion Association, CAS.

## Author contributions

D.-Y.H.: Conceptualization; Writing-original draft; Project adminis-tration; Data curation. D.-B.C.: Conceptualization; Writing-original draft; Methodology; N.-Y.Z.: Data curation; Project administration. Z.-J.W.: Conceptualization. X.-J.H.: Conceptualization. X.L.: Soft-ware; Data curation. M.-Y.L.: Software; Visualization. X.-P.L. Data curation. L.-R.J.: Methodology. J.-P.M. Data curation. T.S.: Super-vision; Validation; Funding acquisition; Writing-review and editing. Z.-Y.Q.: Supervision; Validation; Funding acquisition; Writing-review and editing. W.-H.X.: Supervision; Validation; Funding acquisition; Writing-review and editing. H.W.: Supervision; Validation; Funding acquisition; Writing-review and editing. All authors have read and approved the final version of the manuscript.

## Competing interests

The authors have no competing interest.
