## [Peer review file · Nature Communications]

REVIEWER COMMENTS

Reviewer #1 (Remarks to the Author):

Despite the effectiveness of GPX4 inhibitors in inducing ferroptosis, their application in cancer therapy is hampered by the significant issue of systemic toxicity. In this study, Hou et al. successfully engineered a peptide-ferriporphyrin conjugate (Gi-F-CAA), which is activated specifically within the tumor microenvironment. This innovation enables in situ self-assembly of GPX4 inhibited peptide into nanoparticles that effectively bind with GPX4 to induce ferroptosis in tumor. The author showed significant antitumor efficacy in tumor therapies against bladder tumor, drug-resistant breast tumor and large tumor. Wholly, this rational engineering may inspire more intelligent and precise strategies for tumor therapy. Although it seems very interesting, the detailed biological fate of Gi-F-CAA NPs in cells is unclear. For instance, given the innovative claim on AEB effect, there is no convincing evidence showing that Gi-F-CAA NPs rather than CAA interact with GPX4 in cells. Followings are my specific comments:

1. The author claims the noteworthy assembly enhanced binding (AEB) effect because of the higher binding affinity of Gi-F nanoparticles to GPX4 than the single chain state in Figure 2n. The quantification data of the CAA amount on nanoparticle surface should be described.
2. Please provide the physicochemical properties (e.g., diameter size, zeta potential, morphology, etc.) of nanoparticles, including Gi-F-CAA (pH 6.5), Gi-F, Gi-F-SA and NGi-F-CAA.
3. In the acidic tumor microenvironment, the self-assembled nanoparticles exhibited an increased ability to be internalized through cellular endocytosis. Typically, the lysosome serves as the final compartment in the endocytic pathway. However, since GPX4 locates in cytoplasm and mitochondria, additional experiments and explanations are required to elucidate the subcellular distribution of the self-assembled nanoparticles within cells.
4. There is no convincing data to show whether the NPs escapes from lysosomes.
5. Given the acidic and enzymatic biocontext of lysosomes, will the Gi-F-CAA NPs degrade in lysosomes?
6. The control groups in Figure 3b and 3c should keep consistent.
7. There have been some reported induction manners of ferroptosis by nanoparticles (Nat. Commun., 2020, 11:3484; Biomaterials 2022, 285: 121561; Nanoscale, 2021, 13, 2266). A comparative discussion of these publications with this study will strengthen the implications of this paper.
8. In line 151 page 6, what is the μ represented in "GPX4 μ protein"?

Minor comments:

1. The IHC staining images of GPX4 in Figure 3a are blurry and hazy.
2. The descriptions of scale bar in captions of Figure 3a and 3b are missing.
3. The control group in Figure 3b is inconsistent with its caption.
4. In line 879 page 31, "in vivo" should be corrected as "in vitro".
5. The tumor size in Figure 6f-g may be unethical in Animal ethics committee.

Reviewer #2 (Remarks to the Author):

The manuscript titled "In Vivo Assembly Enhanced Binding Effect Augments Tumor Specific Ferroptosis

Therapy" written by Hou et al. reported a novel chemical entity Gi-F-CAA, which practically is conjugation of a peptide inhibitor of GPX4 (GXpep-3), a PH-sensitive moiety cis-aconitic anhydride (CAA), and ferriporphyrin (FeTPP). Under acidic condition such as tumor microenvironment, the CAA moiety would be hydrolyzed, so the long-chain conjugated molecules would assemble into nanoparticles on its own, because of increased hydrophobicity. The subsequent inhibition of GPX4 activity and enhanced Fenton reaction will lead the tumor cells to over accumulation of lipid hydroperoxide and eventually ferroptosis. As fully demonstrated in the manuscript, such chemical entities appear to accumulate within tumors, and can effectively suppress tumor growth in vivo, in multiple models (bladder tumor, drug-resistant tumor, and large solid tumors).

Overall, I found this strategy to build up conjugated ferroptosis-inducing (FINs) molecule is of outstanding significance, as ferroptosis is an emerging therapeutic approach for cancers, especially those have entered the metastatic phase or have become drug resistant. However, most canonical ferroptosis inducers exhibit poor pharmacokinetic profiles, such as GPX4 inhibitor RSL3. So, the strategy in this manuscript may encourage researchers in the field to try this alternative delivery approach.

To demonstrate the efficacy and mechanism of the reported novel chemical entity, the authors put together a huge amount of work and validated each argument with more than one approach of experiments. I think these approaches/techniques are appropriately used, data quality is more than acceptable, and interpretation and conclusion are reasonable. I am convinced by the efficacy of Gi-F-CAA, but I have difficulties to understand this effect is majorly augmented by Assembly Enhanced Binding (AEB) effect, which is highlighted in the titled. Could it be simply due to accumulation of FINs in tumors? Authors may be able to address this with more demonstrations. I will include my suggestions on strengthening this part in the minor points below.

I would recommend publication of this manuscript with minor revisions. My suggested improvements are listed below:

1. During construction of Gi-F-CAA, an intermediate without iron chelate should be able to isolate. This intermediate can be denoted as TCPP-LKLK(CAA)GACNWLPLYPCPV. I think this intermediate can also be used as a control for Gi-F-CAA, as the difference is with or without Fe. So, the comparison of this pair in cellular or animal models may demonstrate the contribution of Fenton reaction in the observed efficacy. I was wondering if the authors have conducted similar experiments to show this, because all controls seem to have FeTCPP.
2. The author did not present the reason for the selection of GACNWLPLYPCPV as the GPX4 inhibition module in Gi-F-CAA. More explanation on this may be helpful. For instance, why small molecule inhibitors of GPX4 were not used? I suppose small molecules may penetrate tumor better since they are smaller. Besides, GACNWLPLYPCPV was reported before as GXpep-3 in literature. The authors did cite the corresponding literature, but I think a brief mentioning of its origin in introduction may address the curiosity of the audience. And why GXpep-3 is selected over the other two peptidic inhibitors (GXpep-1 and -2)?
3. To demonstrate Assembly Enhanced Binding (AEB) effect, the authors compared K_d of Gi-F-CAA monomer to GPX4 with K_d of Gi-F nanoparticles to GPX4, as measured by MST assay. The unit of K_d is μM or nM . But it is not very clear what the ligand concentrations of Gi-F nanoparticle represents. Is it molar concentration of large nanoparticles, or is it molar concentration of the building blocks (monomers) that were used to generate the nanoparticles? The authors used "apparent K_d " in the writing, but the methods section did not clearly state what is "apparent K_d ". If the concentration of Gi-F refers to number of nanoparticles per volume, it would not be a fair comparison, as nanoparticles will easily saturate the dye-labeled GPX4 and produce a low K_d . Maybe use mass concentration (mg/ml) instead of molar concentration to be clear?

To demonstrate AEB, a control that can be used here in the MST assay is GXpep-3, the inhibitor itself (GACNWLPLYPCPV). It appears that the nanoparticle has higher binding affinity, since GXpep-3 has a K_d of 8.2 μM in the literature. But this depends on what 30.29 nM of Gi-F refers to.

4. To demonstrate "Assembly Enhanced Binding Effect Augments Tumor Specific Ferroptosis Therapy", the authors compared Gi-F-CAA with Gi-F or Gi-F-SA in efficacy study. But if the authors had compared Gi-F-CAA with co-administering of Gi and FeTPP (as free agents, not conjugated), it may be of even higher significance. Because researchers in the field may have done numerous efficacy studies of FINs against tumor, including Gi. If a better efficacy is observed on Gi-CAA over Gi alone, or Gi-F-CAA over Gi + FeTPP (not conjugated, but co-treated), it may encourage more tests on this conjugation strategy. This may be outside the scope of this manuscript. But this manuscript does not have a discussion section to review the limitations and outlooks of this research. The authors may consider adding in more discussions to the end of the manuscript.

Reviewer #3 (Remarks to the Author):

In this manuscript "In Vivo Assembly Enhanced Binding Effect Augments Tumor Specific Ferroptosis Therapy" the authors developed peptide-ferriporphyrin conjugates (Gi-F-CAA) to inhibit GPX4 activity through the assembly enhanced binding (AEB) effect. The authors evaluated the therapeutic effect of Gi-F-CAA and its binding ability with GPX4, highlighting its potential for ferroptosis-based therapy. Furthermore, the authors demonstrated the therapeutic effect of Gi-F-CAA in various in vivo model. Although it is interesting, several results seem unclear and should be revised. Specific several issues should be addressed prior to resubmit the research are listed below:

1. In Figure 2k, the size of Gi-F-CAA binding to GPX4 appears larger than Gi-F-CAA itself. To validate this observation, the authors should use dynamic light scattering (DLS) to determine the size of Gi-F-CAA binding to GPX4.
2. In Figure 3b, the authors assessed the tumor penetrability of Gi-F-CAA in vivo. However, the reason for the deep penetration ability of Gi-F-CAA is not clear. Generally, particles of around 100 nm in size face challenges penetrating the core of tumors due to the extracellular matrix (ECM). Please provide a clear explanation for the enhanced penetration of Gi-F-CAA.
3. In Figure 4, the authors confirmed the therapeutic effect of Gi-F-CAA using various assays. However, it is crucial to determine whether the therapeutic effect of Gi-F-CAA is indeed due to ferroptosis. To demonstrate that the cytotoxicity caused by Gi-F-CAA is specifically related to ferroptosis, the authors should perform an inhibitor assay. This can be achieved by confirming the cell viability after treating the cells with various inhibitors of programmed cell death.
4. In Figure 4d, the authors presented the results showing decreased intracellular GPX4 activity with Gi-F-CAA compared to other groups. However, the difference in GPX4 activity could be influenced by the enhanced cellular uptake of Gi-F-CAA. Therefore, the authors should assess GPX4 activity by mixing only GPX4 and Gi-F-CAA to confirm the direct impact of Gi-F-CAA on GPX4 activity.
5. In Figure 4e, the authors confirmed intracellular Fe^{2+} ions using the FerroOrange probe and provided qualitative results through confocal images. However, it is recommended to include quantitative data to support the increased intracellular iron ions after treatment with Gi-F-CAA.
6. In Figure S13, the TEM image is unclear, and the morphology of Gi-F-CAA is not visible.
7. In Figure S17, the authors confirmed OH radical generation and GSH depletion of Gi-F-CAA and Gi-F. However, the content of reduced GSH was low. Therefore, the authors should investigate the production of OH and depletion of GSH at different concentrations of Gi-F-CAA and Gi-F.

Questions and point-by-point responses are as follows:

Reviewer #1 (Remarks to the Author):

Despite the effectiveness of GPX4 inhibitors in inducing ferroptosis, their application in cancer therapy is hampered by the significant issue of systemic toxicity. In this study, Hou et al. successfully engineered a peptide-ferriporphyrin conjugate (Gi-F-CAA), which is activated specifically within the tumor microenvironment. **This innovation enables in situ self-assembly of GPX4 inhibited peptide into nanoparticles that effectively bind with GPX4 to induce ferroptosis in tumor.** The author showed significant antitumor efficacy in tumor therapies against bladder tumor, drug-resistant breast tumor and large tumor. **Wholly, this rational engineering may inspire more intelligent and precise strategies for tumor therapy.** Although it **seems very interesting**, the detailed biological fate of Gi-F-CAA NPs in cells is unclear. For instance, given the innovative claim on AEB effect, there is no convincing evidence showing that Gi-F-CAA NPs rather than CAA interact with GPX4 in cells. Followings are my specific comments:

1. The author claims the noteworthy assembly enhanced binding (AEB) effect because of the higher binding affinity of Gi-F nanoparticles to GPX4 than the single chain state in Figure 2n. The quantification data of the CAA amount on nanoparticle surface should be described.

Response: Thanks to the reviewer's suggestion. The peptide-ferriporphyrin conjugates (Gi-F-CAA) was consist of a **GPX4** inhibitory peptide (GACNWLPLYPCPV), an assembled peptide linker (LKLKLLK) decorated with a pH-sensitive moiety (cis-aconitic anhydride, **CAA**) and ferriporphyrin (**FeTCPP**). The Gi: **GPX4** inhibitory peptide (GACNWLPLYPCPV) was supposed to on surface of self-assembled nanoparticles interacting with GPX4 in cells after hydrolysis of CAA. We have carefully described the quantification data of the GPX4 inhibitory peptide amount on nanoparticle surface in the revised manuscript.

As expected, the formation of Gi-F nanoparticles was induced by the cleavage of acid-responsive CAA group on the Gi-F-CAA. Thus, the amount of Gi-F monomer in Gi-F nanoparticle could represent the GPX4 inhibitory peptide amount on nanoparticle surface. The quantification amount of Gi-F in the formed nanoparticles was calculated by an approximate method which has reported before (Sci. Adv. 2023, 9, eabq8225; Adv Mater, 2023, doi: 10.1002/adma.202303831). Briefly, as the diameter of nanoparticle is proportional to the molecular weight, according to the reported bovine serum albumin (BSA) protein size and molecular weight (7.2 nm, 66.43 KDa), the molecular weight of Gi-F nanoparticle with an average 78 nm size can be calculated to be 719.66 KDa (Biomacromolecules, 2013, 14, 818–827). The monomer amount Gi-F nanoparticle was calculated as $719.66/2.927 = 246$.

2. Please provide the physicochemical properties (e.g., diameter size, zeta potential, morphology, etc.) of nanoparticles, including Gi-F-CAA (pH 6.5), Gi-F, Gi-F-SA and NGi-F-CAA.

Response: Thanks to the reviewer's suggestion. We have carefully provided the physicochemical properties (e.g., diameter size, zeta potential, morphology, etc.) of nanoparticles, including TEM images of Gi-F (pH 7.4 or pH 6.5) and NGi-F-CAA (pH 6.5); DLS assays of Gi-F (pH 7.4 or pH 6.5), Gi-F-SA (pH 7.4 or pH 6.5) and NGi-F-CAA (pH 6.5); Zeta potential of NGi-F-CAA, Gi-F and Gi-F-SA in the revised manuscript.

Figure 1. (a) Representative TEM images of Gi-F (pH 7.4) and Gi-F (pH 6.5) (40 μ M). Scale bars: 200 nm. (b) Particle size of Gi-F (pH 7.4) and Gi-F (pH 6.5) (40 μ M) measured by DLS.

Figure 2. Particle size of Gi-F-SA (pH 7.4) and Gi-F-SA (pH 6.5) (40 μ M) measured by DLS.

Figure 3. (a) Representative TEM images of NGi-F-CAA (pH 6.5) (40 μ M). Scale bars: 100 nm. (b) Particle size of NGi-F-CAA (pH 7.4) and NGi-F-CAA (pH 6.5) (40 μ M) measured by DLS.

Figure 4. ζ -potential of NGi-F-CAA, Gi-F and Gi-F-SA (pH 7.4) (40 μ M) in PBS. Values are expressed as means \pm S.D. (N = 3).

The manuscript was revised accordingly. The analysis was updated in Figure S13, 14, 15, 16 in the revised Supporting Information.

3. In the acidic tumor microenvironment, the self-assembled nanoparticles exhibited an increased ability to be internalized through cellular endocytosis. Typically, the lysosome serves as the final compartment in the endocytic pathway. However, since GPX4 locates in cytoplasm and mitochondria, additional experiments and explanations are required to elucidate the subcellular distribution of the self-assembled nanoparticles within cells.

Response: Thanks to the reviewer's suggestion. We have carefully performed the colocalization assay between NPs and lysosomes/mitochondria by confocal laser scanning microscopy (CLSM) in the revised manuscript. As a result, NPs were clearly observed inside lysosomes (stained with LysoTracker green) of EJ cells after incubated with EJ cells for 2 h, which demonstrated the endocytosis of NPs. Meanwhile, intensity of red fluorescence signal from NPs increased along with extended incubation time (4 h) whereas it exhibited decreased colocalization with green fluorescence from lysosomes (stained with LysoTracker green), indicating the lysosomes-escape of NPs.

Moreover, NPs were clearly observed to be colocalized with mitochondria (stained with MitoTracker green) at 4 h post-incubation, indicating that lysosomes-escaped NPs might get into cytoplasm and locate in mitochondria.

Figure 5. (a) Representative fluorescence images of EJ cells after incubation with Cy labeled Gi-F-CAA (pH 6.5, 40 μ M) for 2 and 4 h. Lysosomes were labeled with LysoTracker Green. Scale bar: 10 μ m. (b) Representative fluorescence images of EJ cells after incubation with Cy labeled Gi-F-CAA (pH 6.5, 40 μ M) for 4 h. Mitochondria were labeled with MitoTracker Green. Scale bar: 10 μ m.

The manuscript was revised accordingly. The representative fluorescence images were updated in Figure S28 in the revised Supporting Information.

4. There is no convincing data to show whether the NPs escapes from lysosomes.

Response: Thanks to the reviewer's suggestion. We have carefully performed the colocalization assay between NPs and lysosomes by confocal laser scanning microscopy (CLSM) in the revised manuscript. After incubated with EJ cells for 2 h, NPs were clearly observed inside lysosomes

(stained with LysoTracker green) of EJ cells, which demonstrated the endocytosis of NPs. In the meanwhile, intensity of red fluorescence signal from NPs increased along with extended incubation time (4 h) whereas it exhibited poor colocalization with green fluorescence from lysosomes (stained with LysoTracker green), indicating the lysosomes-escape of NPs.

Figure 6. Representative fluorescence images of EJ cells after incubation with Cy labeled Gi-F-CAA (pH 6.5, 40 μ M) for 2 and 4 h. Lysosomes were labeled with LysoTracker Green. Scale bar: 10 μ m.

The manuscript was revised accordingly. The representative fluorescence images were updated in Figure S28a in the revised Supporting Information.

5. Given the acidic and enzymatic biocontext of lysosomes, will the Gi-F-CAA NPs degrade in lysosomes?

Response: Thanks to the reviewer's suggestion. We have carefully performed the HPLC assay to investigate the stability of Gi-F (Gi-F-CAA NPs) in the acidic and enzymatic context of lysosomes in the revised manuscript. As a result, the peak of Gi-F + CTS + LAL (pH 5.0) group was similar to that of Gi-F (pH 7.4) group, indicating the high stability of Gi-F in the acidic and enzymatic context of lysosomes.

Figure 7. HPLC spectra of Gi-F (pH 7.4, 40 μ M) and Gi-F + CTS + LAL (pH 5.0, 40 μ M). CTS: Cathepsin. LAL: lysosomalacidllpase.

The manuscript was revised accordingly. The HPLC spectra was updated in Figure S29 in the revised Supporting Information.

6. The control groups in Figure 3b and 3c should keep consistent.

Response: Thanks to the reviewer's suggestion. The control groups in Figure 3b and 3c was revised to keep consistent.

Figure 8. (a) Fluorescence of images of the whole tumor tissues after treated with Cy labeled Gi-F-CAA, Gi-F and Gi-F-SA (8 mg/kg in 100 μ L PBS). Scale bars: 1 mm. (b) Fluorescence images of tumor tissues after treated with NBD labeled Gi-F-CAA, Gi-F and Gi-F-SA (8 mg/kg in 100 μ L PBS). Scale bars: 20 μ m.

The manuscript was revised accordingly. The fluorescence of images was updated in Figure S24 in the revised Supporting Information.

7. There have been some reported induction manners of ferroptosis by nanoparticles (Nat. Commun., 2020, 11:3484; Biomaterials 2022, 285: 121561; Nanoscale, 2021, 13, 2266). A comparative discussion of these publications with this study will strengthen the implications of this paper.

Response: Thanks to the reviewer's suggestion. We have carefully cited and discussed related works about ferroptosis (Nat Commun, 2020, 11, 3484; Biomaterials, 2022, 285, 121561; Nanoscale, 2021, 13, 2266) in the discussion section of revised manuscript.

8. In line 151 page 6, what is the mu represented in "GPX4mu protein"?

Response: Thanks to the reviewer's suggestion. GACNWLPLYPCPV was reported to inhibit GPX4 in literature (Biochemical and Biophysical Research Communications. 2017, 482, 195e201). To screen random peptide-displaying T7 phage libraries against GPX4, researchers prepared biotinylated Avi-tagged recombinant GPX4 protein possessing seven mutations (GPX4mu). The GPX4mu had a SUMO-tag in its N-terminus and contained substituted amino acid residues (Cys37Ala, Cys93Arg, Cys134Glu, Cys175Val, Cys64Ser, Cys102Ser) for increasing protein solubility. Therefore, the simulation in this work was performed with GPX4mu.

Minor comments:

1. The IHC staining images of GPX4 in Figure 3a are blurry and hazy.

Response: Thanks to the reviewer's suggestion. We have carefully amplified the IHC staining images of GPX4 in Figure 3a in the revised manuscript.

2. The descriptions of scale bar in captions of Figure 3a and 3b are missing.

Response: Thanks to the reviewer's suggestion. We have carefully added the descriptions of scale bar in captions of Figure 3b and 3c in the revised manuscript.

3. The control group in Figure 3b is inconsistent with its caption.

Response: Thanks to the reviewer's suggestion. We have carefully revised that in the revised manuscript.

4. In line 879 page 31, “in vivo” should be corrected as “in vitro”.

Response: Thanks to the reviewer’s suggestion. We have carefully revised that in the revised manuscript.

5. The tumor size in Figure 6f-g may be unethical in Animal ethics committee.

Response: Thanks to the reviewer’s suggestion. According to animal ethics committee, the experimental endpoint in mouse study includes: (1) The tumor burden greater than 10% body weight; (2) The tumor diameter should not exceed 20 mm in any one dimension; (3) The tumor volume does not exceed 2000 mm³. Moreover, we have carefully terminated the in vivo anti-tumor experiment when tumor volume exceeds 2000 mm³ in Figure 6f-g. Therefore, only the last point of the tumor volume exceeds 2000 mm³ at the end of treatment.

Reviewer #2 (Remarks to the Author):

The manuscript titled “In Vivo Assembly Enhanced Binding Effect Augments Tumor Specific Ferroptosis Therapy” written by Hou et al. reported a novel chemical entity Gi-F-CAA, which practically is conjugation of a peptide inhibitor of GPX4 (GXpep-3), a PH-sensitive moiety cis-aconitic anhydride (CAA), and ferriporphyrin (FeTPP). Under acidic condition such as tumor microenvironment, the CAA moiety would be hydrolyzed, so the long-chain conjugated molecules would assemble into nanoparticles on its own, because of increased hydrophobicity. The subsequent inhibition of GPX4 activity and enhanced Fenton reaction will lead the tumor cells to over accumulation of lipid hydroperoxide and eventually ferroptosis. As fully demonstrated in the manuscript, such chemical entities appear to accumulate within tumors, and can effectively suppress tumor growth in vivo, in multiple models (bladder tumor, drug-resistant tumor, and large solid tumors).

Overall, I found this strategy to build up conjugated ferroptosis-inducing (FINs) molecule is of outstanding significance, as ferroptosis is an emerging therapeutic approach for cancers, especially those have entered the metastatic phase or have become drug resistant. However, most canonical ferroptosis inducers exhibit poor pharmacokinetic profiles, such as GPX4 inhibitor RSL3. **So, the strategy in this manuscript may encourage researchers in the field to try this alternative delivery approach.**

To demonstrate the efficacy and mechanism of the reported novel chemical entity, the authors put together a huge amount of work and validated each argument with more than one approach of experiments. **I think these approaches/techniques are appropriately used, data quality is more than acceptable, and interpretation and conclusion are reasonable. I am convinced by the efficacy of Gi-F-CAA**, but I have difficulties to understand this effect is majorly augmented by Assembly Enhanced Binding (AEB) effect, which is highlighted in the titled. Could it be simply due to accumulation of FINs in tumors? Authors may be able to address this with more demonstrations. I will include my suggestions on strengthening this part in the minor points below.

I would recommend publication of this manuscript with minor revisions. My suggested improvements are listed below:

1. During construction of Gi-F-CAA, an intermediate without iron chelate should be able to isolate. This intermediate can be denoted as TCPP-LKLK(LK(CAA)GACNWLPLYPCPV). I think this intermediate can also be used as a control for Gi-F-CAA, as the difference is with or without Fe. So, the comparison of this pair in cellular or animal models may demonstrate the contribution of Fenton reaction in the observed efficacy. I was wondering if the authors have conducted similar experiments to show this, because all controls seem to have FeTCPP.

Response: Thanks to the reviewer’s suggestion. We have carefully used intermediate (Gi-CAA, TCPP-LKLK(LK(CAA)GACNWLPLYPCPV) without iron chelate as a control for Gi-F-CAA in the revised manuscript. As a result, Gi-F-CAA exhibited an obviously enhanced cytotoxicity than that of Gi-CAA based on the cell toxicity assay, demonstrating the contribution of Fenton reaction in the observed efficacy. Meanwhile, the cytotoxicity of Gi-F-CAA was successfully rescued by Fer-1, indicating the ferroptosis-based tumor toxicity.

Figure 9. The viability of EJ cells after treated with Gi-CAA (pH=6.5), Fer-1+Gi-F-CAA (pH=6.5) and Gi-F-CAA (pH=6.5) at different concentrations for 48 h.

The manuscript was revised accordingly. The viability assay was updated in Figure S26 in the revised Supporting Information.

2. The author did not present the reason for the selection of GACNWLPLYPCPV as the GPX4 inhibition module in Gi-F-CAA. More explanation on this may be helpful. For instance, why small molecule inhibitors of GPX4 were not used? I suppose small molecules may penetrate tumor better since they are smaller. Besides, GACNWLPLYPCPV was reported before as GXpep-3 in literature. The authors did cite the corresponding literature, but I think a brief mentioning of its origin in introduction may address the curiosity of the audience. And why GXpep-3 is selected over the other two peptidic inhibitors (GXpep-1 and -2)?

Response: Thanks to the reviewer's suggestion. We have carefully presented the reason for the selection of GACNWLPLYPCPV as the GPX4 inhibition module in Gi-F-CAA rather than small molecule inhibitors of GPX4 in the introduction section of revised manuscript. Both GACNWLPLYPCPV and GPX4 inhibitors (RSL3) was small molecule with advantages of high penetration. However, as a cyclic peptide, GACNWLPLYPCPV possess a higher hydrophilic and longer blood half-life than GPX4 inhibitors (RSL3) with advantages of easily modified.

Meanwhile, a brief mentioning of GACNWLPLYPCPV origin was added in the introduction section of revised manuscript.

Besides, according to corresponding literature, though GXpep-1 (CRVDLQGWRRR), GXpep-2 (CRAWYQNYCALRR) and GXpep-3 (GACNWLPLYPCPV) each peptide exhibited one-to-one binding to GPX4mu and the K_d values were 1.8 μ M, 0.61 μ M and 8.2 μ M, respectively. Unfortunately, GXpep-1 and GXpep-2 did not have an inhibitory effect even at 100 μ M. On the other hand, GXpep-3 exhibited inhibitory activity in a peptide concentration-dependent manner. The IC_{50} value was 10 μ M, which correlated with its K_d value.

Biochemical and Biophysical Research Communications. 2017, 482, 195e201

3. To demonstrate Assembly Enhanced Binding (AEB) effect, the authors compared K_d of Gi-F-CAA monomer to GPX4 with K_d of Gi-F nanoparticles to GPX4, as measured by MST assay. The unit of K_d is μM or nM . But it is not very clear what the ligand concentrations of Gi-F nanoparticle represents. Is it molar concentration of large nanoparticles, or is it molar concentration of the building blocks (monomers) that were used to generate the nanoparticles? The authors used “apparent K_d ” in the writing, but the methods section did not clearly state what is “apparent K_d ”. If the concentration of Gi-F refers to number of nanoparticles per volume, it would not be a fair comparison, as nanoparticles will easily saturate the dye-labeled GPX4 and produce a low K_d . Maybe use mass concentration (mg/ml) instead of molar concentration to be clear? To demonstrate AEB, a control that can be used here in the MST assay is GXpep-3, the inhibitor itself (GACNWLPLYPCPV). It appears that the nanoparticle has higher binding affinity, since GXpep-3 has a K_d of $8.2 \mu\text{M}$ in the literature. But this depends on what 30.29 nM of Gi-F refers to.

Response: Thanks to the reviewer’s suggestion. We have carefully clearly stated “apparent K_d ” in the methods section of revised manuscript. Apparent K_d represented the affinity of assemblies rather than a single molecule. Firstly, the ligand concentrations of Gi-F nanoparticle represent molar concentration of large nanoparticles. The concentration of the Gi-F nanoparticle was calculated by an approximate method which has reported before (Sci. Adv. 2023, 9, eabq8225). Briefly, as the diameter of nanoparticle is proportional to the molecular weight, according to the reported bovine serum albumin (BSA) protein size and molecular weight (7.2 nm, 66.43 KDa), the molecular weight of Gi-F nanoparticle with an average 78 nm size can be calculated to be 719.66 KDa (Biomacromolecules 2013, 14, 818-827). Thus, we could convert the monomer concentration to the Gi-F nanoparticle concentration. As suggested, the

calculation of apparent K_d was added in the method section. Specifically, nanoparticles provide large surface area with exposed GPX4 target to achieve multivalent binding, which could remarkably enhance the binding affinity. This concept has been proven by several papers (Adv. Mater. 2023,35, 2211332; Adv Mater, 2023, doi: 10.1002/adma.202303831). In consistent with the concept, compared with the GXpep-3 ($K_d = 8.2 \mu\text{M}$) which has been reported before (Biochem. Biophys. Res. Commun. 2017, 482, 195-201) and Gi-F-CAA monomer ($K_d = 1.68 \mu\text{M}$), Gi-F nanoparticle exhibit lower binding affinity. The manuscript was revised accordingly.

4. To demonstrate “Assembly Enhanced Binding Effect Augments Tumor Specific Ferroptosis Therapy”, the authors compared Gi-F-CAA with Gi-F or Gi-F-SA in efficacy study. But if the authors had compared Gi-F-CAA with co-administering of Gi and FeTPP (as free agents, not conjugated), it may be of even higher significance. Because researchers in the field may have done numerous efficacy studies of FINs against tumor, including Gi. If a better efficacy is observed on Gi-CAA over Gi alone, or Gi-F-CAA over Gi + FeTPP (not conjugated, but co-treated), it may encourage more tests on this conjugation strategy. This may be outside the scope of this manuscript. But this manuscript does not have a discussion section to review the limitations and outlooks of this research. The authors may consider adding in more discussions to the end of the manuscript.

Response: Thanks to the reviewer’s suggestion. We have carefully compared Gi-F-CAA with co-administering of Gi and FeTPP (as free agents, not conjugated) in the revised manuscript. As a result, Gi-F-CAA exhibited an obviously enhanced cytotoxicity than that of co-administering of Gi and FeTPP based on the cell toxicity assay, indicating the advantages of conjugation strategy.

We have carefully added more discussions of the limitations and outlooks of this research at the end of revised manuscript.

Figure 10. The viability of EJ cells after treated with Gi-CAA (pH=6.5), Fer-1+Gi-F-CAA (pH=6.5) and Gi-F-CAA (pH=6.5) at different concentrations for 48 h.

The manuscript was revised accordingly. The viability assay was updated in Figure S26 in the revised Supporting Information.

Reviewer #3 (Remarks to the Author):

In this manuscript “In Vivo Assembly Enhanced Binding Effect Augments Tumor Specific Ferroptosis Therapy” the authors developed peptide-ferriporphyrin conjugates (Gi-F-CAA) to inhibit GPX4 activity through the assembly enhanced binding (AEB) effect. The authors evaluated the therapeutic effect of Gi-F-CAA and its binding ability with GPX4, highlighting its potential for ferroptosis-based therapy. Furthermore, the authors demonstrated the therapeutic effect of Gi-F-CAA in various in vivo model. Although **it is interesting**, several results seem unclear and should be revised. Specific several issues should be addressed prior to resubmit the research are listed below:

1. In Figure 2k, the size of Gi-F-CAA binding to GPX4 appears larger than Gi-F-CAA itself. To validate this observation, the authors should use dynamic light scattering (DLS) to determine the size of Gi-F-CAA binding to GPX4.

Response: Thanks to the reviewer’s suggestion. We have carefully used dynamic light scattering (DLS) to determine the size of Gi-F-CAA binding to GPX4 in the revised manuscript.

Figure 11. Particle size of Gi-F-CAA (40 μ M) binding to GPX4 in PBS (0.01 M, pH 6.5) measured by DLS.

The manuscript was revised accordingly. The DLS assay were updated in Figure S18 in the revised Supporting Information.

2. In Figure 3b, the authors assessed the tumor penetrability of Gi-F-CAA in vivo. However, the reason for the deep penetration ability of Gi-F-CAA is not clear. Generally, particles of around 100 nm in size face challenges penetrating the core of tumors due to the extracellular matrix (ECM). Please provide a clear explanation for the enhanced penetration of Gi-F-CAA.

Response: Thanks to the reviewer’s suggestion. As previous reported, peptide in single chain state (< 10 nm) possessed a higher penetration ability compared with nanoparticles or antibodies (Angew Chem. 2019, 58, 4632-4637; Nat Commun. 2019, 10, 4861; Adv Mater. 2023. doi: 10.1002/adma.202303831; ACS Appl Mater Interfaces. 2020, 12, 40042-40051). Based on this, Gi-F-CAA was designed to be in single chain state owing to the improved hydrophilicity after CAA modification. Therefore, Gi-F-CAA was supposed to easily penetrate deeply into solid tumor with a small size.

3. In Figure 4, the authors confirmed the therapeutic effect of Gi-F-CAA using various assays. However, it is crucial to determine whether the therapeutic effect of Gi-F-CAA is indeed due to ferroptosis. To demonstrate that the cytotoxicity caused by Gi-F-CAA is specifically related to ferroptosis, the authors should perform an inhibitor assay. This can be achieved by confirming the cell viability after treating the cells with various inhibitors of programmed cell death.

Response: Thanks to the reviewer's suggestion. We carefully performed the cell viability assay with a ferroptosis inhibitor (Fer-1). As a result, Gi-F-CAA exhibited an obviously decreased cytotoxicity after pretreated with Fer-1, demonstrating that the cytotoxicity caused by Gi-F-CAA is specifically related to ferroptosis in the revised manuscript.

Figure 12. The viability of EJ cells after treated with Gi-CAA (pH=6.5), Fer-1+Gi-F-CAA (pH=6.5) and Gi-F-CAA (pH=6.5) at different concentrations for 48 h.

The manuscript was revised accordingly. The viability assay was updated in Figure S26 in the revised Supporting Information.

4. In Figure 4d, the authors presented the results showing decreased intracellular GPX4 activity with Gi-F-CAA compared to other groups. However, the difference in GPX4 activity could be influenced by the enhanced cellular uptake of Gi-F-CAA. Therefore, the authors should assess GPX4 activity by mixing only GPX4 and Gi-F-CAA to confirm the direct impact of Gi-F-CAA on GPX4 activity.

Response: Thanks to the reviewer's suggestion. We have carefully assessed GPX4 activity by mixing only GPX4 and Gi-F-CAA to confirm the direct impact of Gi-F-CAA on GPX4 activity in the revised manuscript.

Figure 13. Glutathione peroxidase 4 (GPX4) activities after treated with PBS (pH=7.4), NGi-F-CAA (pH=7.4), Gi-F-SA (pH=7.4), Gi-F-CAA (pH=7.4) and Gi-F-CAA (pH=6.5, 40 µM). The GPX4 activities of PBS group was normalized as 1.

The manuscript was revised accordingly. The GPX4 activities were updated in Figure S30 in the revised Supporting Information.

5. In Figure 4e, the authors confirmed intracellular Fe²⁺ ions using the FerroOrange probe and provided qualitative results through confocal images. However, it is recommended to include quantitative data to support the increased intracellular iron ions after treatment with Gi-F-CAA.

Response: Thanks to the reviewer's suggestion. We have carefully included quantitative data by Ferrous Ion Content Assay Kit to support the increased intracellular iron ions after treatment

with Gi-F-CAA in the revised manuscript. As a result, the EJ cells treated with Gi-F-CAA (pH 6.5) emitted a substantially increased amount of Fe^{2+} ions compared with that in NGi-F-CAA (pH 7.4), Gi-F-SA (pH 7.4) and Gi-F-CAA (pH 7.4) treated EJ cells.

Figure 14. Intracellular Fe^{2+} ions activities of EJ cells after treated with PBS (pH=7.4), NGi-F-CAA (pH=7.4), Gi-F-SA (pH=7.4), Gi-F-CAA (pH=7.4) and Gi-F-CAA (pH=6.5, 40 μM).

The manuscript was revised accordingly. The quantitative data by Ferrous Ion Content Assay Kit were updated in Figure S32 in the revised Supporting Information.

6. In Figure S13, the TEM image is unclear, and the morphology of Gi-F-CAA is not visible.

Response: Thanks to the reviewer's suggestion. Gi-F-CAA was designed to in single chain state owing to the improved hydrophilicity after CAA modification. On this basis, the Gi-F-CAA possessed the critical aggregation concentrations (CAC) of 149.5 μM , indicating Gi-F-CAA remained as single chains. Moreover, the DLS revealed the sizes of Gi-F-CAA was about 11 ± 2 nm t at pH 7.4. Therefore, the morphology of Gi-F-CAA is not visible in TEM image. The same situation is as follows:

Adv Mater. 2023. doi: 10.1002/adma.202303831

7. In Figure S17, the authors confirmed OH radical generation and GSH depletion of Gi-F-CAA and Gi-F. However, the content of reduced GSH was low. Therefore, the authors should investigate the production of OH and depletion of GSH at different concentrations of Gi-F-CAA and Gi-F.

Response: Thanks to the reviewer's suggestion. We have carefully investigated the production of OH and depletion of GSH at different concentrations of Gi-F-CAA and Gi-F (40 μM or 80 μM) in the revised manuscript.

Figure 15. Relative GSH concentration depletion and •OH generation of Gi-F-CAA and Gi-F (40 µM or 80 µM) after incubation with H₂O₂ (200 µM).

The manuscript was revised accordingly. The GSH depletion and •OH generation were updated in Figure S22 in the revised Supporting Information.

We believe the revised manuscript will meet the wide readership and high impact of *Nature Communications*. We look forward to your response.

REVIEWERS' COMMENTS

Reviewer #1 (Remarks to the Author):

The authors have provided convincing data and addressed all my comments. I recommend to accept this paper.

Reviewer #2 (Remarks to the Author):

After thoroughly reading the response letter of Dr. Hao Wang and his coworkers and the relevant parts of the revised manuscript, I see that they have done a nice job to rework all critical points of my first review. Seemingly, in line with this, they did it for the critical comments of the other reviewers as well. Therefore, I will now suggest the improved manuscript to be accepted for publication at Nature Communications.

Reviewer #3 (Remarks to the Author):

In the revised manuscript, authors addressed all the issues raised by the reviewers and corrected the manuscript accordingly. Therefore, I recommend it to be accepted without further revision.

Reviewer's suggestion and point-by-point responses are as follows:

Reviewer #1 (Remarks to the Author):

The authors have provided convincing data and addressed all my comments. I recommend to accept this paper.

Response: Thanks to the reviewer's approbation.

Reviewer #2 (Remarks to the Author):

After thoroughly reading the response letter of Dr. Hao Wang and his coworkers and the relevant parts of the revised manuscript, I see that they have done a nice job to rework all critical points of my first review. Seemingly, in line with this, they did it for the critical comments of the other reviewers as well. Therefore, I will now suggest the improved manuscript to be accepted for publication at Nature Communications.

Response: Thanks to the reviewer's approbation.

Reviewer #3 (Remarks to the Author):

In the revised manuscript, authors addressed all the issues raised by the reviewers and corrected the manuscript accordingly. Therefore, I recommend it to be accepted without further revision.

Response: Thanks to the reviewer's approbation.

We believe the revised manuscript will meet the wide readership and high impact of *Nature Communications*. We look forward to your response.